# CERTIFIED EVALUATION OF MODEL-LEVEL EXPLANATIONS FOR GRAPH NEURAL NETWORKS

**Sayan Saha**
Machine Intelligence Unit
Indian Statistical Institute, Kolkata
sayansaha181196@gmail.com

**Sanghamitra Bandyopadhyay**
Machine Intelligence Unit
Indian Statistical Institute, Kolkata
sanghami@isical.ac.in

## ABSTRACT

Model-level explanations for Graph Neural Networks (GNNs) aim to identify class-discriminative motifs that capture how a classifier recognizes a target class. Because the true motifs relied on by the classifier are unobservable, most approaches evaluate explanations by their target class score. However, class score alone is not sufficient as high-scoring explanations may be pathological or may fail to reflect the full range of motifs recognized by the classifier. To bridge this gap, this work introduces sufficiency risk as a formal criterion for whether explanations adequately represent the classifier's reasoning, and derives distribution-free certificates that upper-bound this risk. Building on this foundation, three metrics are introduced: Coverage, Greedy Gain Area (GGA), and Overlap which operationalize the certificates to assess sufficiency, efficiency, and redundancy in explanations. To ensure practical utility, finite-sample concentration bounds are developed for these metrics, providing confidence intervals that enable statistically reliable comparison between explainers. Experiments with synthetic data and with three state-of-the-art explainers on four real-world datasets demonstrate that these metrics reveal differences in explanation quality hidden by class scores alone. Designed to complement class score, they constitute the first theoretically certified framework for evaluating model-level explanations of GNNs.

## 1 INTRODUCTION

Graph Neural Networks (GNNs) have emerged as powerful models for learning from graph-structured data, achieving state-of-the-art performance in diverse domains such as drug discovery (Merchant et al., 2023), weather forecasting (Lam et al., 2023) and social network analysis (Wu et al., 2022). Despite these successes, the opaque nature of GNNs pose significant challenges in high-stakes applications, where understanding why a prediction is made is as important as the prediction itself. This has driven rapid progress in research on post-hoc explainability methods for GNNs. Broadly, post-hoc GNN explainability approaches fall into two categories (Kakkad et al., 2023): instance-level and model-level methods. Instance-level methods (Pope et al., 2019; Feng et al., 2023; Baldassarre & Azizpour, 2019; Huang et al., 2022; Schlichtkrull et al., 2020; Yuan et al., 2021; Lucic et al., 2022; Lin et al., 2022; Zhang et al., 2021; Vu & Thai, 2020) are devised to explain the prediction on a single instance by highlighting salient nodes and edges that most influenced the outcome. However, such explanations lack generalizability, since insights drawn from one instance may not extend to others. Hence, achieving a global understanding of model behaviour requires inspecting instance-level explanations across numerous instances, which is costly and unreliable. In contrast, model-level explanations(Yuan et al., 2020; Wang & Shen, 2023; Shin et al., 2024) aim to provide a global view of the classifier's behavior. Given a target class, they seek to identify key discriminative motifs that the model consistently relies on, to recognize instances of the class. These motifs provide a high level view of the classifier's behavior, reducing the need for exhaustive inspection and offering a more coherent picture of what the model has learned.

Model-level explainers typically work by either discovering or generating key class discriminative motifs. While each approach differs in their mechanism, an uniform objective all of them share is to generate motifs that attain high target class scores. Since the true patterns that a classifier relies on to identify a class are unobservable, motifs which the classifier assigns high class scores are in-

terpreted as representative of the discriminative information using which the model has learned to recognize instances of the class. Consequently, the quality of a model-level explanation is typically judged by the score it receives from the classifier, making the target class score the primary metric for comparing explanations and the methods that generate them. However, class score alone is insufficient to distinguish between explanations. Since every explainer explicitly optimizes a loss term that rewards high target class scores, the resulting motifs often become pathological: they achieve high scores but stray far from the data distribution and may bear little resemblance to meaningful graph structures. In the absence of more principled metrics, researchers frequently resort to qualitative inspection where consensus is elusive and comparisons are vulnerable to cherry-picking. Other than qualitative comparison between explanations, researchers also rely on auxilliary measures such as time required to generate explanations, sparsity of the explanations and comparison between graph statistics of real graphs and the generated explanations. While useful, these auxiliary measures do not directly assess explanation quality, since they ignore the relationship between the motifs and the classifier's decision process. It should also be noted here that common measures of explanation quality for instance-level explanations such as fidelity and accuracy style metrics are not directly applicable in this setting. Fidelity style metrics typically involve operations like removing the explanation subgraph or corrupting input features while preserving the explanation. However, model-level explanations, especially those produced by generative methods rarely appear as exact subgraphs of any graph in the class, making such operations infeasible. Accuracy, on the other hand, requires ground-truth explanation subgraphs for comparison. Yet in the model-level setting, the true motif relied upon by the classifier is unknown, rendering this measure inapplicable as well. This leaves a fundamental gap in principled evaluation of model-level explanations.

This work closes this gap by introducing a principled and computable suite of metrics for evaluating model-level explanations. We begin by characterizing when a set of explanations generated by a model-level explainer can be said to sufficiently capture the classifier's decision process, formalizing sufficiency through a risk functional and deriving distribution-free certificates that upper-bound this risk. The first metric, *Coverage*, measures how much of the class manifold in the classifier's embedding space is accounted for by the explanations, thereby providing a certified bound on sufficiency risk. To assess how efficiently coverage is accumulated, we propose the *Greedy Gain Area (GGA)*, which connects to guarantees on prefix coverage, motif budgets, and certified sufficiency under motif constraints, while also diagnosing when coverage has stagnated. We further introduce *Overlap*, which captures redundancy between motifs that explain the same regions. Since these quantities are estimated from finite samples, we also derive uncertainty bounds that yield confidence intervals, ensuring that comparisons between methods remain statistically reliable. Through extensive experiments, we demonstrate that our metrics reliably complement the class score, enabling meaningful distinctions between explanations that would otherwise appear equivalent. Moreover, we show that they can diagnose common pitfalls, such as pathological motifs that drift away from the data distribution and cases of mode collapse where explanations fail to capture the full diversity of patterns learnt by the classifier. To the best of our knowledge, this constitutes the first principled framework with theoretical certificates for evaluating model-level explanations of GNNs.

## 2 RELATED WORK

**Model-Level Explanations.** Compared to instance-level explanations, model-level explanations for GNNs remain relatively underexplored. Existing approaches can be broadly categorized into *generation-based* and *discovery-based* methods. Generation-based methods employ graph generative models to synthesize class-discriminative motifs. Representative examples include XGNN (Yuan et al., 2020), which uses reinforcement learning to construct motifs node by node; GNNInterpreter (Wang & Shen, 2023), which leverages a probabilistic generative model; Graphon-Explainer (Saha & Bandyopadhyay), D4Explainer (Chen et al., 2023), and MAGE (Yu & Gao, 2025), which employ graphon-based, diffusion-based, and motif-based generation strategies respectively. While such methods optimize for high class scores and incorporate distributional constraints, they still yield pathological motifs not representative of the underlying data distribution. Discovery-based methods, such as PAGE and GLGExplainer (Shin et al., 2024; Azzolin et al., 2023), argue that generative explainers are inherently prone to such pathologies and instead identify motifs directly from the dataset, either by selecting discriminative subgraphs already present in instances or by aggregating instance-level explanations. While this avoids unrealistic motifs, there still remains the need

to evaluate whether the extracted patterns generalize across instances in the class, underscoring the need for principled metrics to assess explanation quality.

**Metrics for Evaluating GNN Explanations.** Existing evaluation of GNN explanations combines domain-specific criteria with quantitative metrics such as accuracy, fidelity, faithfulness, sparsity, and consistency. Considerable theoretical analysis has highlighted their limitations and proposed refinements, but all such work remains confined to **instance-level** explanations. Examples include upper bounds on worst-case performance (Agarwal et al., 2022), robust fidelity under distribution shifts (Zheng et al., 2024), characterization scores combining fidelity measures (Amara et al., 2022), and robustness-based metrics (Fang et al., 2023). Complementary benchmarking efforts (Kosan et al., 2024; Agarwal et al., 2023) systematically compare instance-level explainers across diverse datasets and highlight which metrics are reliable in different settings.

Despite this progress, no principled evaluation framework exists for **model-level** explanations. A few works have highlighted the pitfalls of relying solely on class score: Wang & Shen (2023) and Chen et al. (2023) observe that high-scoring explanations may be pathological and drift from the data distribution, while Saha & Bandyopadhyay identify mode collapse, where explainers generate only variations of a single motif despite the classifier relying on diverse patterns. These approaches attempt to mitigate such issues by adding training constraints, but they stop short of introducing metrics that can directly diagnose the issues they highlight.

## 3 PRELIMINARIES

We begin by formalizing when a model-level explainer can be considered to sufficiently capture how a classifier $f(\cdot)$ identifies instances of a target class $c$. Given a target class $c$, a model-level explainer outputs an explanation set $E_c = \{\mathcal{M}_1, \ldots, \mathcal{M}_K\}$, where each motif $\mathcal{M}_i \in E_c$ is intended to represent a discriminative substructure that the classifier relies on to recognize graphs of class $c$.

**Scope of explanation objects.** In this work we use the term **motif** in a broad sense to denote any explanation object that can be mapped to the embedding space of the classifier. This includes extracted subgraphs, prototype instances, generated graphs, or rule instantiated examples produced by model level explainers. The proposed metrics operate only on the embeddings of these objects. Therefore the framework applies uniformly all model -level explainers, as long as their outputs can be embedded by the underlying GNN.

Denote by $\hat{\mathcal{G}}_c$ the set of input graphs that the classifier has identified as belonging to class $c$. Then, the premise of model-level explanation is that for each graph $G \in \hat{\mathcal{G}}_c$, there exists a membership function $M^\star(G, E_c)$ that specifies how motifs in $E_c$ are internally associated by the classifier to $G$ when assigning it to class $c$. In the simplest case, $M^\star(G, E_c)$ may be a binary vector in $\{0, 1\}^K$, where the $i$-th entry indicates whether motif $M_i$ is used in the classifier's reasoning for $G$. More generally, $M^\star(G, E_c)$ may encode soft or probabilistic associations. If $E_c$ truly captures the reasoning of the classifier, then the membership codes $\{M^\star(G, E_c) : G \in \hat{\mathcal{G}}_c\}$ should contain sufficient information to reconstruct the classifier's output scores $\{f_c(G)\}$ for graphs in $\hat{\mathcal{G}}_c$. Note that the true membership function $M^\star$ is not observable, hence in practice a computable surrogate $M$ must be used as an approximation of $M^\star$.

**Notational Simplification:** Since, the explanation set $E_c$ remains fixed throughout our analysis, whenever we denote the input for a membership function $M$, we write $M(G)$ in place of $M(G, E_c)$.

**Sufficiency risk.** Given a membership function $M$ and an explanation set $E_c$, the *sufficiency risk* is defined as

$$SR_c(M, E_c) := \mathbb{E}\left[\left(f_c(G) - \mathbb{E}[f_c(G) \mid M(G)]\right)^2 \;\middle|\; G \in \hat{\mathcal{G}}_c\right],$$

where $f_c(G)$ is the classifier's score for class $c$ and $\mathbb{E}[f_c(G) \mid M(G)]$ its conditional expectation given the membership code. This risk quantifies the predictive information lost when $G$ is replaced by its membership representation under $M$. Low values of $SR_c(M^\star, E_c)$ indicate that motifs faithfully capture the classifier's reasoning, with $SR_c(M^\star, E_c) = 0$ denoting an **optimal** explanation.

## 4 METRICS FOR EVALUATING MODEL-LEVEL EXPLANATIONS

As noted earlier, the true membership $M^\star$ is unobservable, so the true sufficiency risk $SR_c(M^\star, E_c)$ cannot be computed directly (see Appendix J for why we chose this formulation of sufficiency). We therefore consider proxy memberships $M$ induced from the explainer's motifs $E_c$, which approximate $M^\star$ but are inherently noisy. One might ask why we do not simply compute the surrogate $SR_c(M, E_c)$, since both $M(G)$ and $f_c(G)$ are observable. This is feasible in principle as $SR_c(M, E_c)$ is the mean squared error between $f_c(G)$ and its prediction from $M(G)$ using $\mathbb{E}[f_c(G) \mid M(G)]$. However estimating the conditional expectation term requires a high-dimensional regression, which is statistically unstable, sensitive to finite samples, and yields no certified guarantees. A low empirical risk may reflect estimation error rather than genuine sufficiency. For these reasons, we propose computable metrics that serve as distribution-free upper bounds on sufficiency risk, thereby providing reliable certificates of explanatory adequacy.

At this point, it is important to emphasize that, a priori, there is no guarantee that the sufficiency risk under a proxy membership $M$ has any meaningful relation to the risk under the true membership $M^\star$. Without such a connection, bounding $SR_c(M, E_c)$ would say nothing about the unobservable quantity $SR_c(M^\star, E_c)$. The following theorem establishes precisely this link: it shows that the sufficiency risk with any proxy membership is always greater than or equal to that with the true membership. Hence, the risk under $M^\star$ is always upper bounded by the risk under any proxy $M$.

**Theorem 1.** *Fix $E_c$. Let $Y = f_c(G)$. Let $M^\star(G)$ be the (unobservable) true membership relative to $E_c$, and let $M(G) = h(M^\star, \varepsilon)$ be any proxy membership with $\varepsilon \perp Y \mid M^\star$. Then $SR_c(M^\star, E_c) \leq SR_c(M, E_c)$, with equality if and only if $\mathrm{Var}\big(\mathbb{E}[Y \mid M^\star] \mid M\big) = 0$ almost surely.*

The proof is deferred to Appendix B.1. Theorem 1 establishes the ground for introducing a proxy membership function and the first metric Coverage using which the sufficiency risk under the proxy membership can be upper bounded.

### 4.1 COVERAGE

To make sufficiency risk estimable, we introduce a structured proxy membership $M_r$ based on the graph embeddings of the target class and the embeddings of the explanation set $E_c = \{\mathcal{M}_1, \ldots, \mathcal{M}_K\}$. Let $f = H \circ \phi$ where $\phi : \mathcal{G} \to \mathbb{R}^d$ is the classifier's embedding function, and write $m_k := \phi(\mathcal{M}_k)$ for the embedding of each motif. For a graph $G$, define its nearest-motif distance $D(G, E_c) := \min_k \|\phi(G) - m_k\|_2$.

The proxy membership is then the function $M_r : \mathcal{G} \times E_c \to \{1, \ldots, K\} \cup \{\bot\}$

$$M_r(G, E_c) := \begin{cases} \arg\min_k \|\phi(G) - m_k\|_2, & \text{if } D(G, E_c) \leq r, \\ \bot, & \text{otherwise,} \end{cases}$$

where the null code $\bot$ indicates that $G$ is not assigned to any motif. **For clarity, we will simply write $M_r(G)$ and $D(G)$ in the remainder of the paper, with the dependence on $E_c$ left implicit.**

Intuitively, $M_r(G)$ attaches each graph to its closest motif within radius $r$, and leaves it uncovered otherwise. Note that $M_r$ is constructed solely from embeddings and does not use any information beyond what is available to the true membership $M^\star$. Hence, conditioned on the true membership $M^\star$, the residual noise in $M_r$ is independent of $Y$ satisfying the assumption in Theorem 1.

**Definition 1** (**Coverage**). *Coverage is the conditional probability that a class-$c$ instance is covered by some motif.*

$$\mathrm{Cov}_c(r) := \Pr\Big(D(G) \leq r \;\Big|\; G \in \widehat{\mathcal{G}}^c\Big).$$

High coverage means most positive instances for class $c$ are explained by motifs in $E_c$, whereas low coverage indicates explanatory insufficiency.

**Theorem 2** (**Bounding Sufficiency Risk with Coverage**). *Assume the classifier head $H$ in the factorization $f = H \circ \phi$ is $L$-Lipschitz on $\phi(\mathcal{G})$. Then for any $r > 0$, the sufficiency risk under the proxy membership $M_r$ satisfies $SR_c(M_r, E_c) \leq L^2 \mathbb{E}\Big[D(G)^2 \, \mathbf{1}\{D(G) \leq r\} \;\Big|\; G \in \widehat{\mathcal{G}}^c\Big] + \frac{1}{4}\big(1 - \mathrm{Cov}_c(r)\big)$. Moreover, since $D(G)^2 \leq r^2$ whenever $D(G) \leq r$, we obtain the coarser bound $SR_c(M_r, E_c) \leq L^2 r^2 \mathrm{Cov}_c(r) + \frac{1}{4}\big(1 - \mathrm{Cov}_c(r)\big)$.*

The proof is deferred to Appendix B.2. Theorem 2 establishes a computable coverage-based certificate for the sufficiency risk of explanations. A natural question is: for which choice of radius r does this bound become tightest? The following result shows that the optimal choice is the universal radius $r^\star = 1/(2L)$, independent of the distribution of $D(G)$.

**Theorem 3** (**Optimal radius for the coverage certificate**). *Under the setting of Theorem 2, consider the coverage-based bound*

$$B(r) \;=\; L^2 \, \mathbb{E}[D(G)^2 \, \mathbf{1}\{D(G) \leq r\} \mid G \in \widehat{\mathcal{G}}^c] \;+\; \tfrac{1}{4}\,(1 - \mathrm{Cov}_c(r)).$$

*Then $B(r)$ is minimized at $r^\star = \frac{1}{2L}$.*

The proof is deferred to Appendix B.3. The result of Theorem 3 extends verbatim to a scale-invariant, angular formulation on normalized embeddings (Appendix A) which is used in practice compute the Coverage. Together, these results imply that evaluating explanations at radius $r^\star$ yields the tightest certified guarantee on sufficiency risk, regardless of the underlying distribution of distances. The Lipschitz condition required for Theorem 2 is mild, since classifier heads such as linear layers or shallow MLPs with standard activations are Lipschitz and admit efficient spectral-norm based estimates (Xu & Sivaranjani, 2024). While the existence of such an $L$ suffices for the coverage bound, its explicit value becomes relevant only in Theorem 3, where it determines $r^\star$.

## 4.2 GREEDY GAIN AREA: MEASURING DISTRIBUTION OF COVERAGE

Coverage at the optimal radius $r^\star$ provides a certified upper bound on sufficiency risk, but it does not reveal how this coverage is distributed across motifs. An explainer may achieve its coverage by relying almost entirely on a single motif, leaving others redundant, or it may distribute coverage more evenly across motifs. To capture this, we introduce the *Greedy Gain Area (GGA)*.

Formally, given $E_c = \{\mathcal{M}_1, \dots, \mathcal{M}_K\}$, $\phi(\mathcal{M}_i) = m_i$ and radius $r^\star$, define $S_k(r^\star) = \{G \in \widehat{\mathcal{G}}^c : \|\phi(G) - m_k\| \leq r^\star\}$ as the set of class-$c$ graphs covered by motif $\mathcal{M}_k$. We construct a greedy set cover by iteratively selecting motifs: at step $j$, the motif that yields the largest marginal increase in coverage is chosen. Let $I_j \subseteq \{1, \dots, K\}$ denote the set of selected motif indices after $j$ steps. The cumulative coverage fraction after $j$ motifs is then $\alpha_j = \frac{1}{|\widehat{\mathcal{G}}^c|} \left| \bigcup_{k \in I_j} S_k(r^\star) \right|$, $\quad j = 1, \dots, K$.

**Definition 2** (**Greedy Gain Area**). *The* Greedy Gain Area (GGA) *of $E_c$ at radius $r^\star$ is the normalized area under the greedy coverage curve:*

$$\mathrm{GGA}(E_c, r^\star) \;:=\; \frac{1}{K} \sum_{j=1}^{K} \alpha_j.$$

GGA measures how efficiently motifs contribute to coverage: it is high when a few motifs account for most of the explanatory power and low when many motifs are required. Its maximum value equals the coverage attained by the explainer. When coverage is high, a GGA value close to coverage indicates parsimonious explanations where a few motifs suffice, whereas a low GGA reflects diversity with many motifs contributing. When coverage is low, a GGA value close to coverage signals mode collapse, with one motif dominating, while a low GGA denotes poor explanations that are neither sufficient nor diverse.

GGA also admits formal guarantees: it certifies how much coverage is retained when only a fixed budget of motifs is used, and it bounds the resulting sufficiency risk. These guarantees, together with full statements and proofs, are presented in Appendix C.

A different question concerns whether generating additional motifs beyond the given set could still improve explanatory sufficiency. To address this, we examine the greedy coverage curve: once its marginal gains stagnate, further motifs cannot significantly reduce the certified sufficiency risk. In such cases, the explainer can be considered to have already reached its maximal achievable coverage, and producing further motifs yields only marginal benefits. The next theorem formalizes this idea.

**Theorem 4** (Diagnostic stopping criterion from stagnation). *Let $\{\alpha_j\}_{j=1}^{K}$ be the greedy coverage curve at $r^\star$ with marginal gains $\Delta_j = \alpha_j - \alpha_{j-1}$ and final coverage $\alpha^\star = \mathrm{Cov}_c(r^\star)$. Fix $t \in \{1, \dots, K\}$ and suppose the curve stagnates after $t$ motifs in the sense that $\Delta_j \leq \epsilon$ for all $j > t$. Let*

$E_c^{(t)}$ *be the first $t$ motifs under greedy selection and $E_c$ the full set of motifs. Then, under the setting of Theorem 2, $SR_c(M_{r^\star}, E_c^{(t)}) - SR_c(M_{r^\star}, E_c) \leq \frac{1}{4}(\alpha^\star - \alpha_t) \leq \frac{1}{4}(K - t)\epsilon$. In particular, once the marginal gains flatten below $\epsilon$, the certified benefit of adding the remaining motifs is at most $\frac{1}{4}(K - t)\epsilon$. Thus stagnation serves as a diagnostic that the explainer has already achieved essentially all of the coverage it can provide.*

In practice, after generating $K$ motifs, one may inspect the greedy coverage curve. If the curve has already stagnated, Theorem 4 guarantees that producing further motifs cannot yield a meaningful reduction in sufficiency risk. Hence, the curve itself provides a diagnostic criterion for deciding when explanation generation can be safely terminated.

## 4.3 OVERLAP

Even when an explanation set achieves high coverage, different motifs may end up covering largely the same subset of graphs. Such redundancy inflates apparent explanatory capacity without actually broadening the scope of what is explained. To make this effect explicit, we introduce the *Overlap* metric, which measures the degree to which motifs provide duplicated rather than complementary coverage.

**Definition 3** (Overlap). *At radius $r^\star$, define*

$$\text{Overlap} = \frac{\sum_{k=1}^{K} |S_k(r^\star)| - |U(r^\star)|}{\max\{1, |U(r^\star)|\}}, \quad U(r^\star) = \bigcup_{k=1}^{K} S_k(r^\star).$$

*The numerator measures redundant coverage across motifs, while the denominator normalizes by the effective domain size. Overlap takes values in $[0, K-1]$, with $0$ indicating no redundancy and $K - 1$ indicating complete redundancy.*

Taken together, the three metrics provide a holistic evaluation of model-level explanations: coverage certifies sufficiency, GGA characterizes how coverage is accumulated across motifs, and overlap explicitly quantifies redundancy. An analysis of the computational cost of these metrics can be found in Appendix F. We now turn to their empirical estimation, analyzing how reliably these population metrics can be approximated from finite samples.

## 4.4 FINITE-SAMPLE CONCENTRATION OF THE METRICS

The metrics of coverage and greedy gain area are defined as population quantities, i.e., expectations with respect to the distribution of class-$c$ graphs. In practice, however, we only observe a finite sample $\widehat{\mathcal{G}}_c = \{G_1, \ldots, G_n\}$ of size $n$. To make reliable use of such finite data, we derive concentration bounds that quantify the estimation error of the empirical metrics $\widehat{\text{Cov}}_c$ and $\widehat{\text{GGA}}$. These bounds quantify the maximum deviation from the population values at any chosen confidence level $\delta$, effectively yielding confidence intervals whose width shrinks with $n$. Equivalently, they provide confidence intervals that support principled comparison between explainers. Non-overlapping intervals imply statistically distinguishable performance at the population level.

All probability statements below are with respect to the randomness of drawing these $n$ graphs i.i.d. from the conditional distribution given $G \in \widehat{\mathcal{G}}^c$.

**Concentration Bounds for Coverage:** Let, $X_i = \mathbf{1}\{D(G_i) \leq r\}$ be a random variable. At the population level, the true coverage is the expectation $\text{Cov}_c(r) = \mathbb{E}[X_i]$. However, at the empirical level, with $n$ observed graphs, coverage is estimated by the sample average $\widehat{\text{Cov}}_c(r) = \frac{1}{n}\sum_{i=1}^{n} X_i$. The next result quantifies how close this empirical estimate is to its population counterpart.

**Proposition 1** (Concentration of coverage). *For any $\delta \in (0, 1)$, with probability at least $1 - \delta$,*
$$\left|\widehat{\text{Cov}}_c(r) - \text{Cov}_c(r)\right| \leq \sqrt{\frac{1}{2n} \log \frac{2}{\delta}}.$$

**Concentration Bounds for GGA:** As with coverage, GGA can be defined at both the population and empirical levels. At the population level, let $\alpha_j$ denote the expected coverage after selecting $j$ motifs according to the greedy procedure. Then the true GGA is $\text{GGA} = \frac{1}{K}\sum_{j=1}^{K} \alpha_j$.

At the empirical level, with $n$ observed graphs, the coverage after $j$ motifs is given by $\widehat{\alpha}_j$, and the corresponding estimate of GGA is $\widehat{\text{GGA}} = \frac{1}{K} \sum_{j=1}^{K} \widehat{\alpha}_j$.

**Proposition 2** (Concentration of GGA). *For any $\delta \in (0,1)$, with probability at least $1 - \delta$, $\left| \widehat{\text{GGA}} - \text{GGA} \right| \leq \sqrt{\frac{1}{2n} \log \frac{2K}{\delta}}$.*

The proofs of Propositions 1 and 2 are deferred to Appendix B.5 and B.6. Together, Propositions 1–2 establish bounds which translate directly into meaningful confidence intervals. When $n$ is small the intervals are wider, so only coarse distinctions between methods are certified; however, clear performance gaps remain statistically significant. As $n$ increases, the intervals shrink at the rate $O(n^{-1/2})$, enabling progressively finer comparisons between explainers while preserving the same theoretical control. The time complexity bounds for computing the metrics on finite samples data is shown in Appendix F.

## 5 EXPERIMENT

We validate the proposed metrics through controlled experiments designed to highlight key properties of explanations. In all our experiments, the classifier comprises an embedding function followed by a linear classification head, allowing exact computation of its Lipschitz constant and the radius $r^\star$ via the spectral norm. Also note that, Coverage and GGA values in Figs. 1 and 2 and Table 1 are presented with Hoeffding bounds at significance level $p = 0.05$. The datasets and classifiers are discussed in detail in Appendix D.

### 5.1 EXPERIMENTS ON SYNTHETIC DATASET

**Complementing Class Score:** We begin by showing a scenario where the proposed metrics both agree with and differ from class score. We synthetically construct the 4Shapes dataset illustrated in Fig 5 for this purpose. Each class in this dataset corresponds to one of four motifs: Star, Lollipop, Grid or Tree. An instance graph in a class is a random Barabási–Albert (BA) graph to which a motif of the corresponding shape is attached. While graphs in the same class are attached with motifs of the same shape, the number of nodes in the motif and the number of motif attachments vary within a set range. Each class is assigned two explanation sets on which the metrics are computed: a good set containing 5 instances of the class-specific motif without the BA backbone, and a bad set containing 5 random BA graphs. The classifier attains 0.93 test accuracy on this dataset. As shown in Figs. 1a–1b, except for Class 2, good explanation sets always receive both higher class scores and higher coverage. Fig. 1c confirms that good motifs consistently lie closer to the graph embeddings. For Class 2, random BA graphs attain higher class scores but much lower coverage than the good explanation set. This discrepancy indicates that the trigger for Class-2 predictions does not lie near the embedding distribution of Class-2 instance graphs and suggests that the classifier has learned to predict Class 2 whenever it does not see convincing evidence for any of the other three motifs. To confirm this, we input Erdős–Rényi random graphs to the classifier and found that they are also labelled as Class 2 with high confidence. This confirms that absence of the other class-specific motifs is the effective decision rule for Class 2. Considered in isolation, class score makes it appear as though the classifier has learned BA graphs themselves as typical representatives of Class 2, and does not reveal that absence of the other three motifs is driving the prediction. In contrast, the low Coverage of these high-score BA graphs shows that they are off-manifold indicating that the effective decision rule is negative evidence.

**Diagnosing Redundancy and Partial Explanations:** We test whether the metrics capture redundancy in overly similar explanation sets and diagnose when they only partially reflect the motifs recognized by the classifier. We create a synthetic **MixedShapes** dataset by merging pairs of 4Shapes classes. One MixedShapes class contains BA graphs with either a Lollipop or a Star motif attached (never both in the same graph). The other MixedShapes class is defined similarly using Grid and Tree motifs. Thus, each class has two possible motif types, but each graph only has motifs of a single shape. For each MixedShapes class, we build two explanation sets: a *unimodal* set, where all explanations use the same motif type from that class(Star for one class and Grid for another), and a *bimodal* set, where explanations include both motif types. As shown in Fig. 2, while both sets attain similar mean class scores, bimodal sets achieve much higher coverage explaining a broader

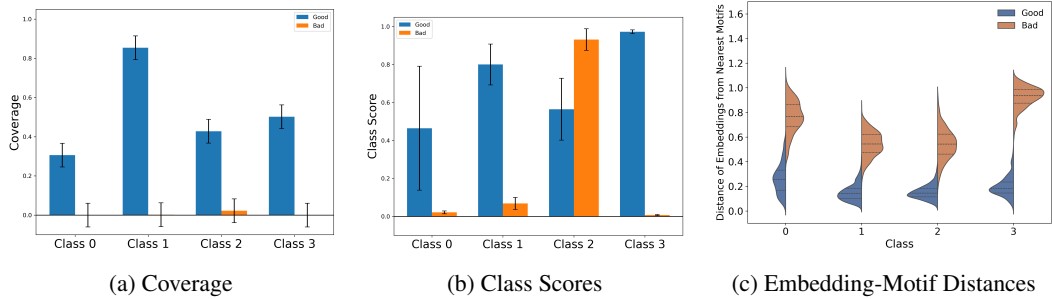

Figure 1: a) Coverage b) Class Scores and c) Distance of Embeddings from Nearest Motifs for Good (**Blue**) and Bad (**Orange**) explanation sets on **4Shapes**.

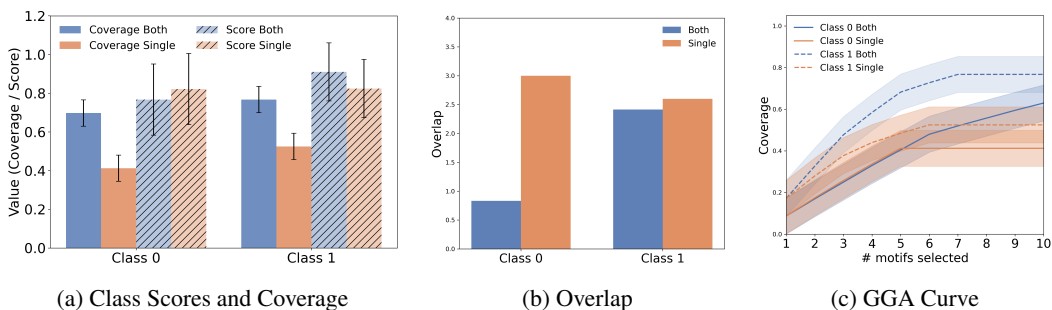

Figure 2: Plots of a) Coverage (**Solid Bars**) and Class Scores (**Dashed Bars**) b) Overlap and c) GGA Curve for **MixedShapes** for Bimodal (**Blue**) and Unimodal (**Orange**) sets.

range of instances, while unimodal sets show high overlap as motifs cover largely the same subset of instances. The GGA curves also show that bimodal motifs steadily expand coverage, whereas unimodal coverage quickly saturates. Thus, the metrics expose sufficiency and redundancy properties invisible to class scores. Additional analysis in Appendix G shows how explanation embeddings align with graph embeddings.

## 5.2 EXPERIMENTS USING STANDARD MODEL-LEVEL EXPLAINERS

We conducted experiments with three standard model-level explainers XGNN (Yuan et al., 2020), GNNInterpreter (Wang & Shen, 2023) and PAGE (Shin et al., 2024) across four diverse real-world datasets: MUTAG (Debnath et al., 1991), IMDB-Multi (Yanardag & Vishwanathan, 2015), REDDIT-Binary (Yanardag & Vishwanathan, 2015), and the OGB-MOLHIV (Hu et al., 2020) dataset.

For each target class, an explanation set of ten motifs is generated by each explainer. The qualitative (Fig. 3) and quantitative (Table 1) results reveal distinct behaviors. On MUTAG, XGNN and PAGE achieve higher Coverage than GNNInterpreter due to their design choices: XGNN enforces valency constraints, and PAGE discovers connected motifs, yielding explanations aligned with molecular structures. GNNInterpreter, lacking such domain-aware constraints, generates disconnected and chemically invalid graphs that attain high class scores but low Coverage. On REDDIT and IMDB, XGNN completely fails(scores reported in Appendix I), producing trivial structures (single nodes or lines Fig. 3) with zero Coverage, GGA, Overlap, and low class scores. XGNN also cannot be run on OGB-MOLHIV, since it only supports graphs with discrete node features.

On REDDIT-Binary, where graphs represent large user interaction networks, GNNInterpreter surpasses PAGE in both Coverage and class score, likely because PAGE's subgraph search fails on large graphs. On IMDB-Multi, both methods achieve high Coverage, GGA, and overlap, indicating that few motifs suffice to explain class identity, though PAGE attains significantly higher class scores.

Figure 3: Explanations and Examples on MUTAG, REDDIT-B and IMDB-Multi datasets

Table 1: Quantitative Results on All Real Datasets

| Methods | Dataset | Class | Coverage | GGA | Overlap | Class Score |
|---|---|---|---|---|---|---|
| **XGNN** | **MUTAG** | Mutagenic | 0.773±0.117 | 0.710±0.150 | 7.314 | 0.966±0.005 |
| | | Non-mutagenic | 0.885±0.185 | 0.829±0.235 | 8.400 | 1.000±0.000 |
| **GNN-Interpreter** | **MUTAG** | Mutagenic | 0.411±0.117 | 0.344±0.150 | 6.573 | 0.987±0.002 |
| | | Non-mutagenic | 0.599±0.185 | 0.514±0.235 | 4.674 | 1.000±0.000 |
| | **Reddit-B** | Class 0 | 0.864±0.051 | 0.629±0.065 | 6.511 | 0.844±0.069 |
| | | TableClass 1 | 0.772±0.038 | 0.529 ±0.048 | 5.361 | 0.975±0.004 |
| | **OGB-MolHIV** | HIV | 0.812±0.117 | 0.773±0.141 | 7.781 | 0.449±0.004 |
| | | non-HIV | 0.431±0.013 | 0.362±0.016 | 7.944 | 0.929±0.021 |
| | **IMDB-Multi** | Class 0 | 0.911±0.018 | 0.893±0.061 | 8.599 | 0.650±0.004 |
| | | Class 1 | 0.956±0.077 | 0.956±0.092 | 9.000 | 0.754±0.000 |
| | | Class 2 | 0.976±0.062 | 0.976±0.074 | 9.000 | 0.711±0.003 |
| **PAGE** | **MUTAG** | Mutagenic | 0.778±0.117 | 0.721±0.150 | 5.611 | 0.992±0.002 |
| | | Non-mutagenic | 0.823±0.185 | 0.639±0.235 | 7.682 | 0.926±0.003 |
| | **Reddit-B** | Class 0 | 0.799±0.051 | 0.681±0.065 | 8.694 | 0.776±0.010 |
| | | Class 1 | 0.575±0.038 | 0.323±0.048 | 6.171 | 0.745±0.009 |
| | **OGB-MolHIV** | HIV | 0.737±0.117 | 0.713±0.141 | 8.944 | 0.578±0.008 |
| | | non-HIV | 0.311±0.013 | 0.215±0.016 | 3.872 | 0.937±0.004 |
| | **IMDB-Multi** | Class 0 | 0.947±0.018 | 0.947±0.061 | 9.000 | 0.955±0.001 |
| | | Class 1 | 0.996±0.077 | 0.996±0.092 | 9.000 | 0.856±0.005 |
| | | Class 2 | 0.966±0.061 | 0.966±0.074 | 9.000 | 0.855±0.004 |

On the large-scale OGB-MOLHIV dataset, none of the methods achieve more than 0.5 Coverage on the majority non-HIV class, and GGA values remain close to Coverage, suggesting stagnation where adding more explanations would not improve coverage. For the minority HIV class, Coverage values are higher but class scores remain low. This stems from the severe class imbalance (∼4%), where the classifier memorizes specific minority molecules rather than learning robust discriminative patterns, leaving the explainers unable to extract motifs that trigger confident predictions. Appendix H supports this interpretation: even removing a single random node from a minority molecule flips its label, indicating that the classifier has memorized examples rather than generalizing.

## 5.3 META-MEASURE ANALYSIS AND GUIDANCE ON PRACTICAL USAGE

**Correlation Analysis and Geometry of Optimality:** To examine whether the proposed metrics behave redundantly, we perform a meta measure analysis by sampling explanation embeddings directly from the classifier latent space. We construct 1000 explanation sets, each containing 10 points drawn

uniformly at random from the normalized embedding space of the trained GNN on the MixedShapes dataset, and treat these points as hypothetical motif embeddings. We compute Coverage, GGA and Overlap for each set using the class specific radius $r^*$. The scatter plots in Fig. 4 show that the three metrics explore a broad feasible region and do not collapse to a single deterministic relationship. By definition, GGA cannot exceed Coverage, and this constraint appears as an upper envelope in the Coverage–GGA plane, but within this envelope the values vary widely. Explanation sets with identical Coverage often exhibit very different GGA values depending on how the gains are distributed across motifs. Overlap behaves almost independently of the other two metrics and increases mainly when motifs influence the same neighbourhoods in the embedding space. These results indicate that the metrics measure distinct geometric properties of explanation sets and are not redundant. The scatter plots further show that there is no inherent tradeoff between the three objectives: explanation sets can simultaneously achieve high Coverage, high GGA and low Overlap, and the feasible region contains many such jointly favourable configurations. This has direct implications for explainer design. Methods that discover in distribution motifs are naturally encouraged toward higher Coverage as seen in Figure 6, while explainers that recover multiple distinct reasoning patterns rather than repeated variants tend to achieve lower Overlap.

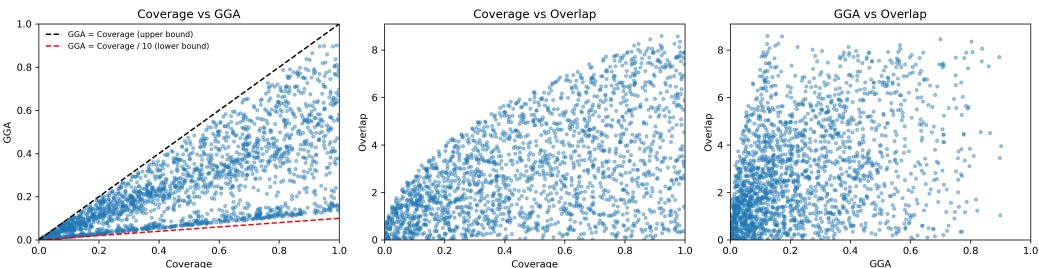

Figure 4: Scatter Plots of Coverage, GGA and Overlap across 1000 explanation sets on the Mixed-Shapes dataset.

**Robustness and Invariance Properties:** The metric system is designed to remain stable under common geometric transformations of the embedding space. Because Coverage, GGA and Overlap are defined entirely through neighborhood relations in the classifier embedding space, they are invariant to uniform feature rescaling, affine gain and other transformations that preserve relative distances. The angular formulation in Appendix A provides an explicit scale-invariant analogue of the Euclidean coverage–risk certificate, and Theorem 5 shows that the optimal angular radius $r_\theta^*$ yields the same ordering of explanation sets as any monotone transformation of the underlying distance. Consequently, replacing Euclidean distance by angular distance or normalizing embeddings does not alter the qualitative conclusions about explanation sufficiency, efficiency or redundancy. Empirically, as shown in Appendix E, the ordering between explanation sets is preserved across small perturbations of the radius, and the metrics vary smoothly under shifts in the embedding geometry. Further, when exact calculation of the Lipschitz constant $L$ is not possible such as for an MLP, it can be upper bounded by multiplying the spectral norms of all weight matrices in the MLP. This provides a computable estimate $\widehat{L} \geq L$ and the sufficiency bounds remain valid for the radius estimated from $\widehat{L}$. Only the tightness of the bound may vary under varying radius.

## 6 CONCLUSION

In this paper, we have examined the limitations of prevailing practices for evaluating model-level explanations in GNNs, highlighting the inadequacy of solely using class score. Motivated by the goal of upper bounding sufficiency risk, we introduced three complementary metrics-Coverage, GGA and Overlap that capture essential properties of explanation sets and are supported by strong theoretical guarantees. Through extensive experiments, we demonstrated that these metrics reliably diagnose unfaithfulness, redundancy, and mode collapse, which remain hidden when evaluation relies only on class score. Our contributions advance the foundations of explainable graph learning by providing the first rigorous theoretically grounded evaluation protocol for model-level explanations.

ACKNOWLEDGEMENTS

The authors were supported in part through the J.C. Bose Grant JBR/2021/000036 from ANRF, Government of India.

**Reproducibility Statement** The code of our project for computing the metrics can be found at `https://github.com/amisayan/Metrics-for-Model-Level-Explanations-of-GNNs`. The proofs of all proposed theorems are in Appendix B. XGNN, GNNInterpreter and PAGE were run using their official implementations which are publicly available. The datasets and the classifiers used on each dataset is detailed in Appendix D.

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

## A  COVERAGE IN ANGULAR DISTANCE

**Why angular distances?**  In practice, the Euclidean geometry underlying $r_E^\star = \frac{1}{2L}$ can be fragile to arbitrary rescalings of the embedding space or the classifier head (e.g., layerwise gain, feature normalization choices), which stretch or shrink $\|\phi(G) - m_k\|_2$ without altering the decision boundary. As a result, the Euclidean coverage at $r_E^\star$ may saturate to 0 or 1 and become uninformative, despite the validity of the theoretical guarantee. To obtain a *scale-invariant* notion of proximity while retaining a distribution-free upper bound on sufficiency risk, we evaluate coverage on *normalized* embeddings using *angular* distance. The next result provides the exact analogue of the Euclidean bound and its optimizer in this scale-invariant geometry.

**Theorem 5** (Coverage–risk bound and optimal radius in angular geometry). *Let $f = H \circ \phi$ and assume the head $H$ is $L$-Lipschitz w.r.t. the Euclidean norm on $\phi(\mathcal{G})$: $|H(z_1) - H(z_2)| \leq L\|z_1 - z_2\|_2$ for all $z_1, z_2 \in \phi(\mathcal{G})$. Define normalized embeddings $\tilde{\phi}(G) = \phi(G)/\|\phi(G)\|_2$ and $\tilde{m}_k = m_k/\|m_k\|_2$, and the angular distance*

$$D_\theta(G) := \min_k \theta\big(\tilde{\phi}(G), \tilde{m}_k\big), \qquad \theta(u, v) := \arccos\big(\langle u, v\rangle\big).$$

*Let $\mathrm{Cov}_c^\theta(r_\theta) := \Pr\big(D_\theta(G) \leq r_\theta \mid G \in \widehat{\mathcal{G}}^c\big)$. Then for any $r_\theta \in (0, \pi]$,*

$$SR_c(M_{r_\theta}, E_c) \leq L^2 \, \mathbb{E}\Big[\big(2\sin(\tfrac{1}{2}D_\theta(G))\big)^2 \mathbf{1}\{D_\theta(G) \leq r_\theta\} \,\Big|\, G \in \widehat{\mathcal{G}}^c\Big] + \tfrac{1}{4}\big(1 - \mathrm{Cov}_c^\theta(r_\theta)\big), \quad (1)$$

*and the right-hand side is minimized at*

$$r_\theta^\star = 2 \arcsin\Big(\min\{1, \tfrac{1}{4L}\}\Big).$$

*When $L \geq \frac{1}{4}$, this simplifies to $r_\theta^\star = 2\arcsin(\frac{1}{4L})$.*

*Proof.* Define the proxy $M_{r_\theta} \in \{1, \ldots, K, \perp\}$ by assigning $G$ to the index of its nearest motif in angle if $D_\theta(G) \leq r_\theta$, and to $\perp$ otherwise. Let $Y := f_c(G) \in [0, 1]$. By variance decomposition,

$$SR_c(M_{r_\theta}, E_c) = \sum_{k=1}^{K} \Pr(M_{r_\theta} = k)\,\mathrm{Var}(Y \mid M_{r_\theta} = k) + \Pr(M_{r_\theta} = \perp)\,\mathrm{Var}(Y \mid M_{r_\theta} = \perp).$$

For a covered bin ($M_{r_\theta} = k$), $\mathrm{Var}(Y \mid M_{r_\theta} = k) \leq \mathbb{E}[(Y - f_c(\mathcal{M}_k))^2 \mid M_{r_\theta} = k]$. On the unit sphere, the chord–angle identity yields $\|\tilde{z} - \tilde{m}\|_2 = 2\sin\big(\tfrac{1}{2}\theta(\tilde{z}, \tilde{m})\big)$. Using Lipschitzness of $H$ in Euclidean norm,

$$|f_c(\tilde{\phi}(G)) - f_c(\tilde{m}_k)| \leq L\,\|\tilde{\phi}(G) - \tilde{m}_k\|_2 = 2L\sin\big(\tfrac{1}{2}\theta(\tilde{\phi}(G), \tilde{m}_k)\big).$$

Averaging over covered bins gives the expectation term in equation 1. For the uncovered bin, since $Y \in [0, 1]$, $\mathrm{Var}(Y \mid M_{r_\theta} = \perp) \leq \frac{1}{4}$, giving the second term.

For optimality, let $F_\theta(r) := \Pr(D_\theta(G) \leq r \mid G \in \widehat{\mathcal{G}}^c)$ and write the bound as

$$B_\theta(r_\theta) = L^2 \int_{[0, r_\theta]} 4\sin^2(\tfrac{t}{2})\, dF_\theta(t) + \tfrac{1}{4}\big(1 - F_\theta(r_\theta)\big).$$

For $0 \leq r_1 < r_2 \leq \pi$,

$$B_\theta(r_2) - B_\theta(r_1) = \int_{(r_1,r_2]} \left(4L^2 \sin^2(\tfrac{t}{2}) - \tfrac{1}{4}\right) dF_\theta(t).$$

Hence $B_\theta$ decreases while $4L^2 \sin^2(r_\theta/2) \leq \tfrac{1}{4}$ and increases afterwards. The minimizer solves $2\sin(r_\theta^\star/2) = \tfrac{1}{2L}$. If $L \geq \tfrac{1}{4}$, this yields $r_\theta^\star = 2\arcsin(\tfrac{1}{4L})$; otherwise the inequality holds for all $t \in [0, \pi]$ and the minimizer is the maximal feasible radius. Compactly, $r_\theta^\star = 2\arcsin(\min\{1, \tfrac{1}{4L}\})$. $\square$

# B PROOFS

## B.1 PROOF OF THEOREM 1

*Proof.* First note that for any $M$, $SR_c(M, E_c) = \mathbb{E}\left[\mathrm{Var}(Y \mid M)\right]$. Set $Z := \mathbb{E}[Y \mid M^\star]$. By the tower property of expectation,

$$\mathbb{E}[Y \mid M] = \mathbb{E}\left[\mathbb{E}[Y \mid M^\star] \mid M\right] = \mathbb{E}[Z \mid M].$$

By the law of total variance applied inside $M$,

$$\mathrm{Var}(Y \mid M) = \mathbb{E}\left[\mathrm{Var}(Y \mid M^\star) \mid M\right] + \mathrm{Var}(Z \mid M) \geq \mathbb{E}\left[\mathrm{Var}(Y \mid M^\star) \mid M\right].$$

Taking expectations yields

$$SR_c(M, E_c) = \mathbb{E}\left[\mathrm{Var}(Y \mid M)\right] \geq \mathbb{E}\left[\mathrm{Var}(Y \mid M^\star)\right] = SR_c(M^\star, E_c),$$

as claimed. $\square$

## B.2 PROOF OF THEOREM 2

*Proof.* Let $Y = f_c(G)$. By variance decomposition,

$$SR_c(M_r, E_c) = \mathbb{E}[\mathrm{Var}(Y \mid M_r(G))] = \sum_{k=1}^{K} \Pr(M_r = k)\,\mathrm{Var}(Y \mid M_r = k) + \Pr(M_r = \perp)\,\mathrm{Var}(Y \mid M_r = \perp).$$

We show the total bound by bounding the covered and uncovered parts.

**Covered Part ($M_r = k$).** For each $k$, let $\mathcal{M}_k \in E_c$ denote the $k$-th motif with embedding $m_k = \phi(\mathcal{M}_k)$. For any random variable $Z$ and constant $a$, $\mathrm{Var}(Z) \leq \mathbb{E}[(Z-a)^2]$. Apply this with $Z = Y$ and $a = f_c(\mathcal{M}_k)$ under the conditional distribution given $M_r = k$:

$$\mathrm{Var}(Y \mid M_r = k) \leq \mathbb{E}\left[\left(f_c(G) - f_c(\mathcal{M}_k)\right)^2 \mid M_r = k\right].$$

By $L$-Lipschitzness of $H$ on $\phi(\mathcal{G})$,

$$|f_c(G) - f_c(\mathcal{M}_k)| \leq L\,\|\phi(G) - m_k\|_2,$$

hence

$$\mathrm{Var}(Y \mid M_r = k) \leq L^2\,\mathbb{E}\left[\|\phi(G) - m_k\|_2^2 \mid M_r = k\right].$$

Averaging over $k$ with weights $\Pr(M_r = k)$ yields

$$\sum_{k=1}^{K} \Pr(M_r = k)\,\mathrm{Var}(Y \mid M_r = k) \leq L^2\,\mathbb{E}[D(G)^2\,\mathbf{1}\{D(G) \leq r\}].$$

**Uncovered Part ($M_r = \perp$).** Here $M_r$ takes a single value, so

$$\Pr(M_r = \perp)\,\mathrm{Var}(Y \mid M_r = \perp) = \left(1 - \mathrm{Cov}_c(r)\right)\mathrm{Var}(Y \mid M_r = \perp).$$

Since $Y = f_c(G) \in [0, 1]$, any conditional variance satisfies $\mathrm{Var}(Y \mid \cdot) \leq 1/4$. Thus

$$\Pr(M_r = \perp)\,\mathrm{Var}(Y \mid M_r = \perp) \leq \tfrac{1}{4}\left(1 - \mathrm{Cov}_c(r)\right).$$

Combining the covered and uncovered parts establishes the claim. $\square$

### B.3 PROOF OF THEOREM 3

*Proof.* Let $F(r) = \Pr(D(G) \leq r \mid G \in \widehat{\mathcal{G}}^c)$ denote the cumulative distribution function of $D(G)$. For two radii $0 \leq r_1 < r_2$, the change in the bound is

$$B(r_2) - B(r_1) = L^2 \int_{(r_1, r_2]} t^2 \, dF(t) \; - \; \tfrac{1}{4}\big(F(r_2) - F(r_1)\big).$$

This expression rewrites as

$$B(r_2) - B(r_1) = \int_{(r_1, r_2]} \big(L^2 t^2 - \tfrac{1}{4}\big) \, dF(t).$$

We now analyze the sign of the integrand: - If $r_2 \leq 1/(2L)$, then for all $t \in (r_1, r_2]$ we have $L^2 t^2 \leq \tfrac{1}{4}$. Hence each term inside the integral is nonpositive, so $B(r_2) - B(r_1) \leq 0$. Therefore $B(r)$ is nonincreasing on $[0, 1/(2L)]$. - If $r_1 \geq 1/(2L)$, then for all $t \in (r_1, r_2]$ we have $L^2 t^2 \geq \tfrac{1}{4}$. Hence each term inside the integral is nonnegative, so $B(r_2) - B(r_1) \geq 0$. Therefore $B(r)$ is nondecreasing on $[1/(2L), \infty)$.

Combining these two facts, the function $B(r)$ decreases up to $r = 1/(2L)$ and increases thereafter. Thus $B(r)$ achieves its global minimum at

$$r^\star = \tfrac{1}{2L}.$$

$\square$

### B.4 PROOF OF THEOREM 4

*Proof.* All expectations/probabilities are with respect to $G \in \widehat{\mathcal{G}}^c$. By Theorem 2, for any explanation set $E$ at $r^\star$,

$$SR_c(M_{r^\star}, E) \; \leq \; L^2 \, \mathbb{E}\big[D_E(G)^2 \, \mathbf{1}\{D_E(G) \leq r^\star\}\big] + \tfrac{1}{4}\big(1 - \alpha_E\big),$$

where $D_E(G)$ is the nearest-motif distance to $E$ and $\alpha_E$ its coverage fraction. For the greedy prefix $E_c^{(t)}$ and the full set $E_c$, enlarging the motif set cannot increase the distance term, i.e., $D_{E_c}(G) \leq D_{E_c^{(t)}}(G)$ and hence

$$\mathbb{E}\big[D_{E_c}(G)^2 \, \mathbf{1}\{D_{E_c}(G) \leq r^\star\}\big] \; \leq \; \mathbb{E}\Big[D_{E_c^{(t)}}(G)^2 \, \mathbf{1}\{D_{E_c^{(t)}}(G) \leq r^\star\}\Big].$$

Therefore

$$SR_c(M_{r^\star}, E_c^{(t)}) - SR_c(M_{r^\star}, E_c) \; \leq \; \tfrac{1}{4}\big(\alpha^\star - \alpha_t\big).$$

If $\Delta_j \leq \epsilon$ for all $j > t$, then $\alpha^\star - \alpha_t = \sum_{j=t+1}^{K} \Delta_j \leq (K - t)\epsilon$, yielding the stated bound. $\square$

### B.5 PROOF OF PROPOSITION 1

*Proof.* By Hoeffding's inequality for bounded independent variables in $[0, 1]$, for any $t > 0$,

$$\Pr\big(|\widehat{\mathrm{Cov}}_c(r) - \mathrm{Cov}_c(r)| \geq t\big) \leq 2\exp(-2nt^2).$$

Set the right-hand side to $\delta$ and solve for $t$: $2\exp(-2nt^2) = \delta \iff t = \sqrt{\tfrac{1}{2n} \log \tfrac{2}{\delta}}$. Equivalently, $\Pr\big(|\widehat{\mathrm{Cov}}_c(r) - \mathrm{Cov}_c(r)| \leq \sqrt{\tfrac{1}{2n} \log \tfrac{2}{\delta}}\big) \geq 1 - \delta$. $\square$

### B.6 PROOF OF PROPOSITION 2

*Proof.* Fix $j$. Write $\widehat{\alpha}_j = \tfrac{1}{n} \sum_{i=1}^{n} X_{i,j}$ with indicators $X_{i,j} \in [0, 1]$. Hoeffding gives $\Pr(|\widehat{\alpha}_j - \alpha_j| \geq t) \leq 2\exp(-2nt^2)$ for any $t > 0$. Apply a union bound over $j = 1, \ldots, K$:

$$\Pr\Big(\max_{1 \leq j \leq K} |\widehat{\alpha}_j - \alpha_j| \geq t\Big) \leq 2K \exp(-2nt^2).$$

Set the RHS to $\delta$ and solve for $t$: $2K \exp(-2nt^2) = \delta \iff t = \sqrt{\tfrac{1}{2n} \log \tfrac{2K}{\delta}}$. Finally,

$$\big|\widehat{\mathrm{GGA}} - \mathrm{GGA}\big| = \Big|\tfrac{1}{K} \sum_{j=1}^{K} (\widehat{\alpha}_j - \alpha_j)\Big| \leq \max_j |\widehat{\alpha}_j - \alpha_j|$$

which yields the stated bound with probability at least $1 - \delta$. $\square$

## C  ADDITIONAL GUARANTEES USING GGA

When only a budget of $t < K$ motifs are retained for interpretability, can we certify how much coverage is still guaranteed? The following theorem shows that GGA provides exactly such a guarantee: it lower bounds the prefix coverage $\alpha_t$ at every step $t$, using only the achievable coverage $\mathrm{Cov}_c(r^\star)$ and the average area summarized by GGA.

**Theorem 6** (Prefix coverage guarantee from GGA). *Let $\alpha^\star = \mathrm{Cov}_c(r^\star)$ be the total achievable coverage at $r^\star$. Then for every $t \in \{1, \ldots, K\}$, the coverage obtained by the first $t$ greedy motifs is certified to satisfy*

$$\alpha_t \;\geq\; K\,\mathrm{GGA} \;-\; (K - t)\,\alpha^\star.$$

*Thus GGA provides a distribution-free lower bound on prefix coverage, linking overall efficiency to budgeted motif selection.*

*Proof.* We have

$$\sum_{j=1}^{K} \alpha_j = \sum_{j=1}^{K} \sum_{i=1}^{j} \Delta_i = \sum_{i=1}^{K} (K - i + 1)\,\Delta_i.$$

Split the sum at $t$:

$$\sum_{j=1}^{K} \alpha_j = \sum_{i=1}^{t} (K - i + 1)\Delta_i + \sum_{i=t+1}^{K} (K - i + 1)\Delta_i.$$

For $i \leq t$, $(K - i + 1) \geq (K - t + 1)$, and for $i > t$, $(K - i + 1) \leq (K - t)$. Hence

$$K\,\mathrm{GGA} = \sum_{j=1}^{K} \alpha_j \leq (K - t + 1)\alpha_t + (K - t)(\alpha^\star - \alpha_t).$$

Rearranging gives $\alpha_t \geq K\,\mathrm{GGA} - (K - t)\alpha^\star$. $\qquad\square$

Its immediate consequences: bounding the number of motifs required to reach a target coverage fraction and the corresponding sufficiency risk of such prefixes are presented next.

**Corollary 1** (Motif budget for target fraction of coverage). *Fix $p \in (0, 1]$ and let $T_p = \min\{j : \alpha_j \geq p\,\alpha^\star\}$ be the number of motifs required to reach $p$ fraction of the achievable coverage $\alpha^\star$. Then*

$$T_p \;\leq\; \left\lceil K + p \;-\; K\,\frac{\mathrm{GGA}}{\alpha^\star} \right\rceil.$$

*Proof.* By Theorem 6, for any $t$ we have

$$\alpha_t \;\geq\; K\,\mathrm{GGA} - (K - t)\alpha^\star.$$

If $K\,\mathrm{GGA} - (K - t)\alpha^\star \geq p\,\alpha^\star$, then $\alpha_t \geq p\,\alpha^\star$ and hence $T_p \leq t$. Rearranging yields $t \geq K + p - K(\mathrm{GGA}/\alpha^\star)$. Taking the smallest integer $t$ that satisfies this gives the bound. $\quad\square$

**Corollary 2** (Risk bound under motif budget). *Let $E_c^{(t)}$ denote the greedy prefix of size $t$ at $r^\star$, with coverage $\alpha_t$. Then by Theorem 2, the sufficiency risk satisfies*

$$SR_c(M_{r^\star}, E_c^{(t)}) \;\leq\; L^2\,\mathbb{E}[D(G)^2\,\mathbf{1}\{D(G) \leq r^\star\}] + \tfrac{1}{4}(1 - \alpha_t).$$

*Using Theorem 6, this implies the certified bound*

$$SR_c(M_{r^\star}, E_c^{(t)}) \;\leq\; L^2\,\mathbb{E}[D(G)^2\,\mathbf{1}\{D(G) \leq r^\star\}] + \tfrac{1}{4}\Big(1 - \max\{0, K\,\mathrm{GGA} - (K - t)\alpha^\star\}\Big).$$

## D  DETAILED EXPERIMENTAL SETUP

All experiments were conducted using the PyTorch Geometric library (Fey & Lenssen, 2019) for implementing and training the graph neural networks, and the NetworkX library (Hagberg et al., 2007) for synthetic graph generation and visualization. The experiments were performed on a high-performance workstation equipped with an Intel Xeon Processor with 40 cores, 256 GB RAM, and an NVIDIA Quadro RTX 6000 GPU (24 GB).

---

**Algorithm 1 4Shapes** Dataset Generation

---

**Require:** Number of classes $C = 4$, number of graphs per class $N$, range of motifs per graph $[m_{\min}, m_{\max}]$, number of nodes in each motif $[n_{\min}, n_{max}]$
**Ensure:** Synthetic graph dataset $\mathcal{D}$
1: **for** each class $c \in \{0, 1, 2, 3\}$ **do**
2:     **for** $i = 1$ to $N$ **do**
3:         Generate a Barabási–Albert (BA) graph as the backbone
4:         Select motif type based on class $c$:
                •    Class 0: Star
                •    Class 1: Grid
                •    Class 2: Lollipop
                •    Class 3: Balanced tree
5:         Sample $m \sim \mathcal{U}[m_{\min}, m_{\max}]$
6:         Sample $n \sim \mathcal{U}[n_{\min}, n_{\max}]$
7:         **for** $j = 1$ to $m$ **do**
8:             Generate a motif of the chosen type with $n$ nodes
9:             Attach the motif to the backbone by connecting a random node in the backbone with a random node in the motif
10:        **end for**
11:        Add the resulting graph with label $c$ to $\mathcal{D}$
12:     **end for**
13: **end for**
14: **return** $\mathcal{D}$

---

## D.1 DATASETS

**4Shapes.** We construct this synthetic dataset which consists of 4000 graphs divided into four classes with 1000 graphs each. Each graph is generated by first constructing a Barabási–Albert (BA) backbone. Depending on the class label, one of four motif types (star, grid, lollipop, or balanced tree) is selected. A random number of motifs is sampled and attached to the backbone by connecting a random node from the motif to a random node in the backbone. Algorithm 1 demonstrates the generation algorithm of the dataset and Figure 5 shows representative examples of instances in the dataset and good and bad explanation sets.

**MixedShapes.** This synthetic dataset is built from **4Shapes** by merging pairs of classes. The first MixedShapes class combines the Star and Lollipop classes: each graph is a BA graph with either a Star or a Lollipop motif attached (never both in the same graph). The second MixedShapes class is defined similarly by merging the Grid and Tree classes, so each graph has either a Grid or a Tree motif. Thus, each class has two possible motif types, but every graph contains exactly one motif of a single shape.

**MUTAG.** MUTAG (Debnath et al., 1991) is a benchmark dataset of 188 molecular graphs, each labeled according to the mutagenic effect of the compound on *Salmonella typhimurium*. In these graphs, nodes denote atoms (e.g., C, O, N) and edges represent chemical bonds. Node features typically encode atom types, while edge features capture bond types. Although relatively small, MUTAG is widely used for evaluating graph classification methods and explanation approaches, as models often highlight substructures such as functional groups that influence mutagenicity.

**REDDIT-B.** REDDIT-B (Yanardag & Vishwanathan, 2015) is a large-scale social network dataset containing 2,000 graphs constructed from Reddit discussion threads. Each graph corresponds to a thread, with nodes representing users and edges indicating reply interactions. The task is binary classification: Q&A communities, which yield tree-like structures, versus discussion communities, which form denser interaction patterns. With graphs often containing hundreds of nodes and no node features, REDDIT-B is commonly used to benchmark GNN scalability and their ability to capture structural information from purely relational data.

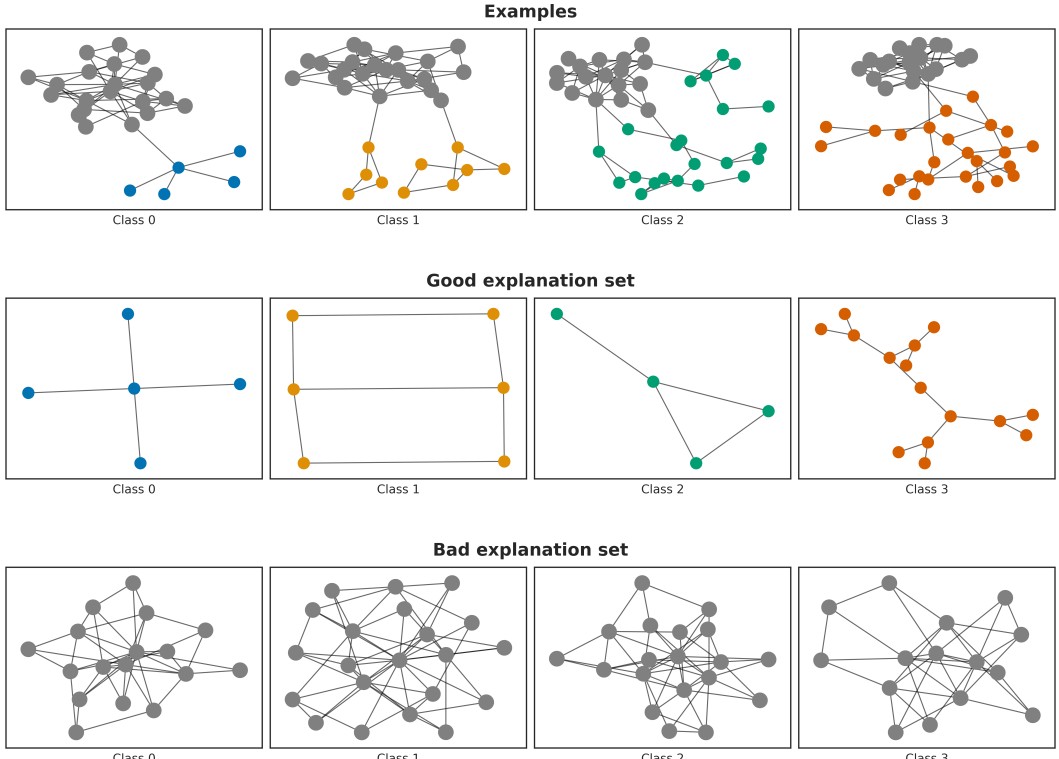

Figure 5: Representative Examples of Instance Graphs in the 4Shapes dataset with Examples in the Good and Bad Explanation Sets. The Good explanation sets shows examples of the motifs contained in each class while the bad explanation sets are random BA graphs.

**IMDB-MULTI.** IMDB-MULTI (Yanardag & Vishwanathan, 2015) consists of 1,500 ego-networks extracted from the Internet Movie Database. Each graph corresponds to an actor/actress, with nodes as actors and edges connecting pairs who co-appear in movies. Graphs are labeled into three classes based on the predominant genre of the target actor's movies: Action, Comedy, or Drama. These graphs are sparse and lack node features, making the dataset a standard benchmark for evaluating GNN performance on social networks with limited attribute information.

**OGBG-MOLHIV.** OGBG-MOLHIV, part of the Open Graph Benchmark (OGB)(Hu et al., 2020), contains 41,127 molecular graphs with the binary prediction task of determining whether a molecule inhibits HIV replication. Rich atom-level (type, chirality, valence) and bond-level features are provided. Dataset splits are defined using scaffold splitting, ensuring structurally distinct molecules across train, validation, and test sets. As a large-scale and chemically meaningful benchmark, ogbg-molhiv is widely adopted for testing the generalization ability of GNNs in molecular property prediction and drug discovery applications.

### D.2 CLASSIFIER DETAILS

**Model architecture and Training** For the IMDB-MULTI, REDDIT-B and OGB datasets, we employ a Graph Isomorphism Network (GIN) architecture. The encoder consists of five stacked GIN-Conv layers, each parameterized by a two-layer MLP with ReLU activation. Batch normalization and dropout ($p = 0.5$) are applied after each layer to improve stability and prevent overfitting. The final node embeddings are aggregated using global mean pooling to obtain a fixed-size graph-level representation. This pooled representation is passed through a fully connected layer with ReLU activation, followed by a linear classifier that outputs logits over the class labels. The model is trained using the Adam optimizer with a learning rate of $10^{-3}$ and a cross-entropy loss function. A StepLR scheduler with decay factor $\gamma = 0.5$ is applied every 20 epochs to reduce the learning rate adaptively.

On the MUTAG dataset the general architecture and the training paradigm remains the same with the only modification being the use of two stacked GINConv layers.

On the 4Shapes and MixedShapes datasets, we use a simple GNN classifier consisting of two GCN-Conv layers with 64 hidden units and ReLU activations, followed by global sum pooling and a final fully connected linear layer that outputs the class logits.

Table 2: Dataset Properties and Classifier Accuracy

| Dataset | #Classes | #Graphs | Average #Nodes | Average #Edges | Classifier Accuracy |
|---|---|---|---|---|---|
| **IMDB-Multi** | 3 | 1500 | 19.77 | 96.53 | 0.835 |
| **Reddit-Binary** | 2 | 2000 | 429.63 | 497.75 | 0.871 |
| **MUTAG** | 2 | 188 | 17.93 | 19.79 | 0.8723 |
| **OGB-MOLHIV** | 2 | 41127 | 25.51 | 54.94 | 0.9701 |
| **4Shapes** | 4 | 2000 | 41.53 | 103.20 | 0.9320 |
| **MixedShapes** | 2 | 2000 | 41.53 | 103.20 | 0.9563 |

## E  BEHAVIOUR OF COVERAGE: ALIGNMENT WITH DISTRIBUTIONAL DISTANCE AND ROBUSTNESS TO RADIUS CHANGES

**Relationship with Wasserstein Distance:** To verify that coverage reflects distributional fidelity, we compare it against the Wasserstein (W1) distance between explanation sets and the target class distribution. We generate 1000 explanation sets on the **4Shapes** dataset for each class and group them into quantiles based on coverage. Figure 6 shows violin plots of W1 distances for each quantile. The pattern is clear: sets with higher coverage exhibit substantially lower and more concentrated W1 values, while low-coverage sets show large and dispersed distances. This confirms that coverage is not only a sufficiency certificate but also correlates with being *in-distribution*: explanations that align more closely with the class distribution naturally achieve higher coverage.

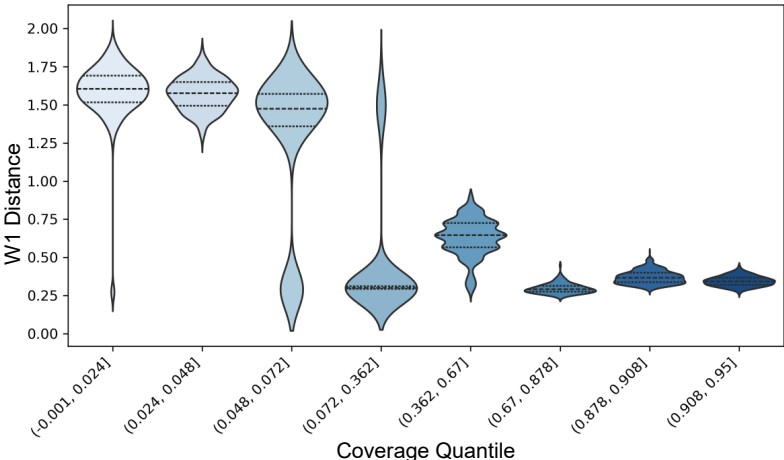

Figure 6: Distribution of W1 distances across coverage quantiles (1,000 explanation sets). Higher coverage corresponds to lower and less variable W1, indicating stronger alignment with the class distribution.

**Robustness to Radii Changes:** We further test robustness to the choice of radius $r$ on which the coverage is evaluated. For this coverage is evaluated over a sweep of radii around $r^\star$, comparing good and random motif sets on the **4Shapes** dataset. As shown in Figure 7, across all radii, the ordering remains invariant: good motifs consistently achieve higher coverage than random ones, with no reversals observed. This stability shows that coverage-based comparisons are robust to radius selection.

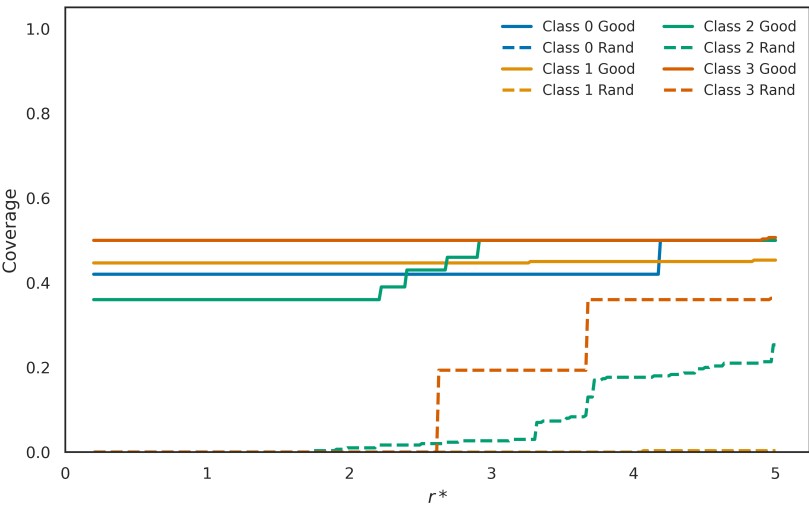

Figure 7: Coverage variance with $r^\star$

## F  COMPUTATIONAL COMPLEXITY OF METRICS

We analyze the computational cost of computing the proposed metrics. Let $N$ denote the number of positive embeddings and $M$ the number of motif embeddings considered for a class.

| Metric | Complexity | Notes |
|---|---|---|
| Coverage | $O(NM)$ | Distance matrix + minimum operation |
| GGA | $O(NM^2)$ | Greedy selection of motifs |
| Overlap | $O(NM)$ | Computed from coverage sets |

Table 3: Computational complexity of the proposed metrics.

**Coverage.**   Coverage requires computing the minimum distance from each of the $N$ embeddings to the $M$ motifs. This involves constructing an $N \times M$ distance matrix and a minimum operation, leading to a complexity of
$$O(NM).$$

**GGA (Greedy Gain Area).**   The GGA curve is obtained by iteratively adding motifs that maximize marginal coverage. At each iteration, coverage gains for all remaining motifs are evaluated, costing $O(NM)$ per iteration. Over up to $K$ iterations (with $K \leq M$), this yields a worst-case complexity of
$$O(NM^2).$$

**Overlap.**   Overlap is computed from the coverage sets constructed during GGA. Since these sets are already available, the additional computation is linear in the number of memberships, giving
$$O(NM).$$

**Summary.**   The complexities of the three metrics are summarized in Table 3. In practice, the number of motifs $M$ is small ( 5–10), so the quadratic dependence in GGA remains computationally feasible.

## G  FURTHER ANALYSIS ON THE **MIXEDSHAPES** DATASET

We compare the explanation sets constructed from *Both-shapes motifs* (bimodal, containing both motif types per class) and *Single-shape motifs* (unimodal, containing only one motif type per class).

The t-SNE visualization in Figure 8 a shows that the bimodal explanation set achieves a broader and more even coverage of the embedding space, with motifs distributed across diverse regions. In contrast, the unimodal set concentrates motifs in fewer regions, leaving large portions of the embedding space underrepresented.

The nearest-distance distributions in Figure 8 b confirm this observation. Embeddings tend to lie closer to their nearest motifs under the bimodal set, whereas the unimodal set results in larger distances, indicating poorer representational coverage. Together, these results highlight that explanation sets combining multiple motif types provide better alignment with the embedding distribution and are thus more representative.

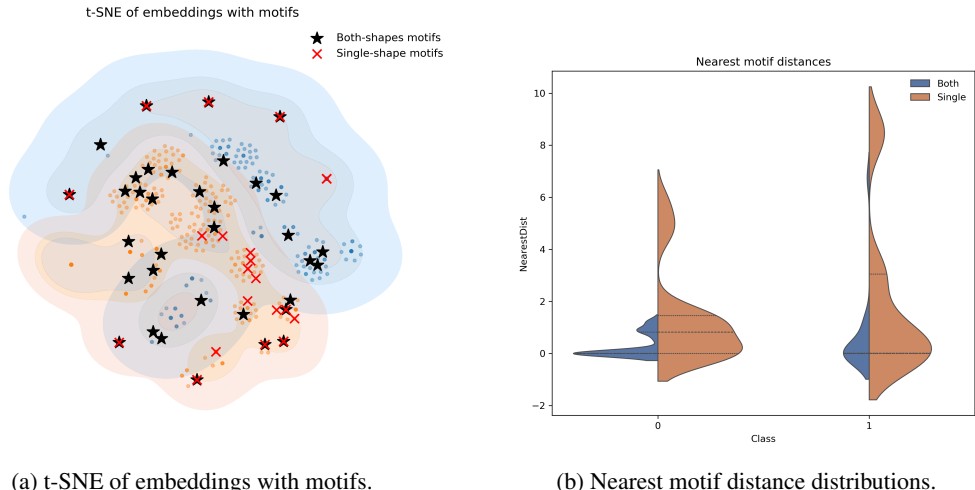

(a) t-SNE of embeddings with motifs.    (b) Nearest motif distance distributions.

Figure 8: Comparison of bimodal (Both-shapes) vs unimodal (Single-shape) explanation sets on the **MixedShapes** dataset.

# H ANALYSIS ON THE OGB DATASET

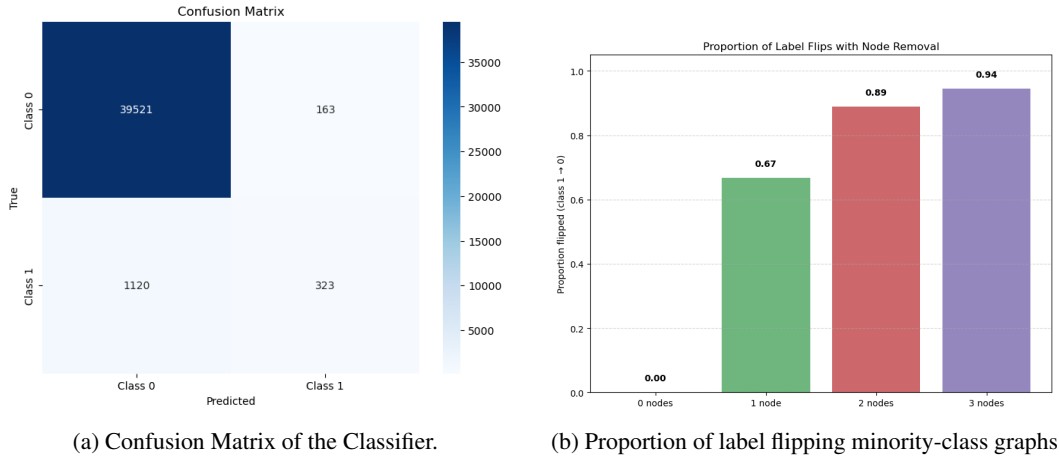

(a) Confusion Matrix of the Classifier.   (b) Proportion of label flipping minority-class graphs

Figure 9: Classifier Analysis on the OGB-Molhiv Dataset

Figure 9a shows the confusion matrix of the classifier trained on the OGB dataset. It can be observed that the classifier has a high rate of misclassification on the minority class. We saw that the explainers on the minority class struggled to achieve a high class score on this dataset even though they attained relatively high Coverage scores. Our hypothesis for this phenomenon was that the classifier has memorized instances it has classified to the minority class rather than learning a general pattern.

This is why the no discriminative motif that the explainers came up with attained a high class score. We verify our hypothesis using a controlled experiment. We randomly choose 50 graphs that the classifier has classified to the minority class and progressively delete 1-3 randomly chosen nodes from each graph. Figure 9b shows the proportion of graphs whose labels flip upon deletion of nodes. It can be seen that after deletion of one node 67% of the graphs flip labels to the majority class while after deletion of three nodes 94% of the graphs flip their labels. This shows the strong bias of the classifier to the majority class. Since, the node deletion is done in random and the chosen graphs were also random, it shows that the classifier's decision is not robust to minor perturbation. In other words, the classifier has not learnt a general pattern for the minority class. It has only memorized certain instances that it has rightly or wrongly classified to the minority class.

## I    RESULTS ON IMDB-MULTI AND REDDIT-B FOR XGNN

Table 4 shows the quantitative results of XGNN on the IMDB-Multi and the REDDIT-Binary datasets.

Table 4

| Methods | Dataset | Class | Coverage | GGA | Overlap | Class Score |
|---------|---------|-------|----------|-----|---------|-------------|
| XGNN | Reddit-B | Class 0 | 0.000±0.051 | 0.000±0.065 | 0.000 | 0.003±0.000 |
| | | Class 1 | 0.000± 0.038 | 0.000 ± 0.048 | 0.000 | 0.357±0.000 |
| | IMDB-Multi | Class 0 | 0.000±0.018 | 0.000±0.061 | 0.000 | 0.499± 0.000 |
| | | Class 1 | 0.000±0.077 | 0.000±0.092 | 0.000 | 0.544±0.000 |
| | | Class 2 | 0.000±0.062 | 0.000±0.074 | 0.000 | 0.508±0.000 |

## J    MOTIVATION FOR THE SUFFICIENCY RISK FORMULATION

**Sufficiency as a sufficient statistic.**    Our notion of *sufficiency* is grounded in the classical concept of a *sufficient statistic*. A statistic $T(X)$ is sufficient for predicting a target quantity if, once $T(X)$ is known, the original variable $X$ contains no additional predictive information about that target. In our setting, the graph $G$ is a random variable and the membership code $M(G)$ is a statistic of $G$. The goal is to quantify how far $M(G)$ is from being sufficient for predicting the classifier score $f_c(G)$.

**Why the conditional expectation is the canonical predictor under squared loss.**    Consider predicting $f_c(G)$ using only the information in $M(G)$, i.e., predictors of the form $g(M(G))$. A standard projection (orthogonality) result states that the conditional expectation $\mathbb{E}[f_c(G) \mid M(G)]$ is the unique minimizer of mean squared error over this function class:

$$\mathbb{E}[f_c(G) \mid M(G)] \;=\; \arg\min_g \; \mathbb{E}\big[(f_c(G) - g(M(G)))^2\big]. \tag{2}$$

Equivalently, for any measurable $g$,

$$\mathbb{E}\big[(f_c(G) - g(M(G)))^2\big] = \underbrace{\mathbb{E}\big[(f_c(G) - \mathbb{E}[f_c(G) \mid M(G)])^2\big]}_{\text{sufficiency risk}} + \mathbb{E}\big[(\mathbb{E}[f_c(G) \mid M(G)] - g(M(G)))^2\big].$$

$$\tag{3}$$

Thus, any squared-loss notion of "how sufficient is $M(G)$ for predicting $f_c(G)$" must be measured relative to this optimal predictor.

**Interpretation via law of total variance.**    By the law of total variance, the sufficiency risk admits the equivalent form

$$\mathcal{R}_{\text{suf}}(M) \;:=\; \mathbb{E}\big[(f_c(G) - \mathbb{E}[f_c(G) \mid M(G)])^2\big] = \mathbb{E}[\text{Var}(f_c(G) \mid M(G))]. \tag{4}$$

Hence $\mathcal{R}_{\text{suf}}(M) = 0$ if and only if $f_c(G)$ is (almost surely) a deterministic function of $M(G)$, i.e., $M(G)$ is sufficient for predicting $f_c(G)$ in the standard statistical sense.

**Relation to our certificates.** While one could consider information-theoretic alternatives (e.g., mutual information between $G$ and $M(G)$), our goal is to relate an ideal membership $M^\star$ to a proxy membership $M_r$ and derive distribution-free guarantees. The conditional-expectation formulation in equation 2–equation 4 connects directly to our Lipschitz-based bounds, enabling certificates for the gap in sufficiency risk between $M^\star$ and the proxy.

## K   LLM USAGE

We used GPT-5 to polish the writing and grammar of the paper.

