# OpenReview forum: "Certified Evaluation of Model-Level Explanations for Graph Neural Networks"
_ICLR.cc/2026/Conference — ICLR 2026 Poster_

### Official Review · Reviewer_GqAd · 2025-10-17

**Soundness:** 2
**Presentation:** 1
**Contribution:** 3
**Rating:** 2
**Confidence:** 4

**Summary:**

In short, the paper proposes a new way of evaluating model-level explanations.

More in detail, the paper introduces sufficiency risk as a metric that equals zero for optimal explanations. Then, since the metric is difficult to compute, the paper introduces practical metrics, like Coverage, Greedy Gain Area, and Overlap. Coverage, in particular, is shown to upper-bound sufficiency risk, making it a principled way to estimate the risk. Convergence and approximation bounds from finite samples are further discussed, showing how the metrics can be efficiently and reliably estimated from limited data.

**Strengths:**

- I agree that there is a fundamental lack in evaluation metrics for model-level explainers, and the paper proposes to fill the gap with an interesting suite of metrics. I think this contribution fills a big gap in the current literature.

- I also agree that the usual approach of evaluating the classifier score metric cannot capture all the nuances of the explanation. The paper describes this issue in detail and motivates well why the literature should move beyond that.

- The proposed metrics are principled, and the authors show how to reliably estimate their values, and also show how metrics complement each other.


This paper **has the potential to be a cornerstone paper** in the context of model-level explainer, and I liked a lot the motivation part. Nonetheless, **when it comes to the results and implementation side, I'm left with some concerns**, detailed below.

**Weaknesses:**

- **W0**: The authors say that estimating sufficiency risk is particularly challenging, as it involves a conditional expectation. However, I struggle to understand why the provided formulation is the only possible way to define the sufficiency risk. Do you really need to formulate it as a conditional expectation? I think more details should be provided about the rationale of this formulation. Also, the authors write:

> Low values of $SR_c(.)$ indicate that motifs faithfully capture the classifier’s reasoning.

However, the authors claim that class core alone is not sufficient to distinguish good from bad explanations; therefore, **how can sufficiency risk alone be indicative of an explanation's faithfulness, given that it is an MSE between class scores?**


- **W1**: Authors define the proxy membership function $M_r$ as a proximity-based check detailed in lines 188-191. However, I feel that a natural baseline for this would be to run some subgraph isomorphism check, which, albeit more computationally expensive, does not require defining a hyperparameter $r^*$ and is not approximated. I was expecting the authors to at least compare their proposed methodology with that simple baseline, to show it is actually better/more efficient than such a baseline.

- **W2**: Line 196 says *the residual noise in $M_r$ is independent of $Y$ satisfying the assumption in Theorem 1*. However, I believe that considering the embeddings given by $\phi$ does not make $M_r$ independent of $Y$, as $Y$ is still predicted from $\phi$. So, I believe there is still a correlation between embeddings and classifier scores. Actually, embeddings contain all the information needed by the classifier to predict $Y$, making it clearly non-uncorrelated with $Y$.

- **W3:** In lines 105-106, the authors write: "*While this avoids unrealistic motifs, extracted patterns often cover only a limited subset of instances and may fail to generalize across the class*". Since this is a strong statement showing key limitations of discovery-based approaches, some evidence or literature in support of the claim should be provided.

- **W4:** Greedy Gain Area seems to favor explanations where few motifs cover a large portion of the input graphs (*it is high when a few motifs account for most of the explanatory power and low when many motifs are required*). Nonetheless, I believe that the opposite should be actually preferred: If only a few motifs cover the majority of graphs, then the other motifs present in the explanation are redundant, therefore indicating an explanation with useless motifs.

- **W5:** The authors consider only a very simple model with a linear classification head in the experiments, allowing exact computation of $r*$. It is not clear how these results generalize when the exact computation of $r*$ is not possible, and how the approximation of $r*$ for more complex models will impact the metric values.

- **W6:** It is unclear from the text that the 4Shapes dataset is actually proposed by the authors. Also, several implementational details are missing in the main text, like a proposed description of how motifs look like (their name is not very indicative), the test accuracy reached by the trained GNN, and the specific architecture of that GNN. Some of these details are reported in the Appendix, but no explicit link to that is presented in the main text, making it hard to parse the details coherently.

- **W7:** The sentence in line 349:

> For evaluation, each class is assigned two explanation sets: a good set containing 5 instances of the class-specific motif without the BA backbone and a bad set containing 5 random BA graphs.

is not clear. What do you mean by evaluation here? Also, why does the good set of explanations contain 5 instances of the class-specific motif instead of just 1? Aren't these motifs always the same for a fixed class?

- **W8:** In line 354, the authors say:

> As shown in Figs. 1a–1b, good sets achieve both high class scores and coverage, while random BA graphs may reach high class scores but always yield zero coverage.

is not true. In fact, Fig. 1a shows indeed very low coverage results, but for Class 2, the score is greater than zero.


- **W9:** In line 353, the authors claim that

>  random BA graphs may reach high class scores

as in Fig. 1b class scores for Class 2 are considerably high, and this is presented as a weakness of the class score metric previously adopted by other papers. Nonetheless, **I feel this does not necessarily show the intended failure case**. In fact, a valid strategy for the GNN to solve the 4Shapes task is to recognize three out of four motifs, and predict the fourth class when such three motifs are not there in the graph. I suspect this is what is happening in Class 2 in the figure. Therefore, an explanation highlighting a random BA graph is actually a good explanation for Class 2, as it indeed does not contain any of the three other motifs, and this would invalidate your claim that the Class Score can be erroneously high for *bad* explanations.


- **W10:** In line 377, the authors write:

> For each class, we compare a unimodal explanation set (containing a single motif)

but it is unclear which single motif they are referring to, as classes have 2 possible motifs appearing together. In general, this experimental part should be better described and formalized more clearly.


- **W11:** Authors provide a link to an anonymous GitHub repo, but the link is provided as a hyperref, which stops people reading the paper on paper from actually seeing the link. Also, I checked the repo, and **the files *metrics.py* and *requirements.txt* are empty**.

**Minors**

- **M1:** In line 51, the authors write that a uniform objective of model-level explainers is to generate motifs attaining a high target class score. Nonetheless, alternative formulations exist, like Azzolin et al. 2023 (already cited), and [1] which I think should be added to the discussion.

- **M2:** Citation format is not consistent across the paper. I would suggest sticking to *\citep{}* for non-in-text citations.

- **M3:** Some theoretical results, like Theorem 2, are introduced abruptly. A more gentle and intuitive description would be better.

- **M4:** Line 228, in particular the statement *admit efficient spectral-norm based estimates*, needs some references in support.

- **M5:** Authors use the word *classifier score* and *class score* interchangably, which creates confusion. I would suggest sticking to only one way of referring to that concept.





[1] GraphTrail: Translating {GNN} Predictions into Human-Interpretable Logical Rules. NeurIPS 2024.



**In summary**, I believe the paper in the current format is not solid enough for acceptance, but I remain open to reconsidering the score if the authors improve it.

**Questions:**

- Q1: I'm expecting that $SR_c(M*, E_c)$ to be always zero, as $M^*$ is the true membership function. Is this correct?

- Q2: How is the membership function $M$ defined for disconnected motifs appearing in the same sample for a certain class, like *square AND lollipop*? And how does the proposed proxy membership function $M_r$ cope with that situation? Examples of such explanations can be found in Azzolin et al. 2023 and [1].

- Q3: Fig 1c shows that the embedding-motif distances separate quite well good from bad explanations. Hence, why not use that as a metric by itself, by setting a threshold on the distance to separate good from bad explanations?

- Q4: Do the authors think that evaluating a model-level explanation by feeding the classifier with only the individual motif may cause some OOD issues in the evaluation, so that model's prediction changes not because the motif is not representative of what the model is doing, but just because the input sample is OOD?

---

> ### Author Response · Authors · 2025-11-23
> **Response to Reviewer GqAd**
>
> We thank you for engaging with our work in extensive detail. We are happy that you found our contribution as filling a **big gap** in the model-level literature. Finally, we are delighted that you think that the paper has the potential to be a **cornerstone paper** in this domain. We have tried to polish and revise the paper (**revised part is in blue for ease of reference**)  to address your comments to achieve this potential.  Please find our response to your comments pointwise below:
>
> **W0.** Our notion of sufficiency is grounded in the classical concept of a sufficient statistic. A statistic $T(X)$ is sufficient for predicting a target if, once $T(X)$ is known, the original variable $X$ contains no additional predictive information about that target. In our setting, the graph $G$ is the random variable and the membership code $M(G)$ is a statistic of $G$. The sufficiency risk is designed to quantify how far this statistic is from being sufficient for predicting the classifier’s output $f_c(G)$.
>
> The conditional expectation enters through a standard result in statistics which states that among all measurable functions $g(M(G))$, the conditional expectation $g(M(G)) = \mathbb{E}[f_c(G)\mid M(G)]$ is the unique function that minimizes the squared error $\mathbb{E}[(f_c(G) - g(M(G)))^2]$. In other words, it is the best possible predictor of $f_c(G)$ that can be constructed using only the information in the statistic $M(G)$. By the law of total variance, our sufficiency risk is exactly the expected conditional variance $\mathbb{E}[\mathrm{Var}(f_c(G)\mid M(G))]$, which is zero if and only if $f_c(G)$ is a deterministic function of $M(G)$. This is precisely the sense in which ``$M(G)$ is sufficient for $f_c(G)$''. We do not claim this is the only possible definition, but it is a canonical one.
>
> We did consider information--theoretic alternatives such as mutual information between $G$ and $M(G)$, but this did not give us a clear way to relate the ideal membership $M^\star$ and the proxy $M$, which is central to our analysis. In contrast, the conditional-expectation formulation leads directly to the Lipschitz-based bounds we derive for the gap between the sufficiency risk of $M^\star$ and that of a proxy.
>
> Regarding the second point, we agree that simple class-score heuristics alone cannot distinguish good from bad explanations; this is exactly the critique we make of prior work. However, sufficiency risk is not just MSE between class scores. It is a population quantity that depends on how scores behave over the entire distribution of graphs that the classifier has labelled as class $c$, not just on the explanation motifs themselves. For example, a motif may achieve a high class score on its own, but the sufficiency risk will be low only if the classifier consistently uses the discriminative information represented by that motif across the real instances it assigns to class $c$ (i.e., scores are stable within each membership group). If scores fluctuate widely within the same membership pattern, the conditional variance $\mathrm{Var}(f_c(G)\mid M(G))$ is large and the sufficiency risk is high, even though the raw class score on the motif is high.
>
> Finally, we reiterate, our goal is not to replace class score but to complement it. We believe understanding a classifier's behaviour at the model-level has two parts to it. One is understanding what triggers the classifier to behave in a certain way which is the component captured by Class Score and another is how the classifier behaves on the distribution of real instances which is captured by the metrics proposed in the paper.
>
> **W1.**  We did consider using a subgraph-isomorphism or graph-matching algorithm initially to define the proxy membership function. However, there are two primary reasons why we ultimately did not adopt this. While formally defining subgraph isomorphism between two graphs is possible, it is only really meaningful when explanations are literal subgraphs of instance graphs, which is not the case in our setting, especially for generative model-level explainers whose outputs are not necessarily subgraphs of any instance graph, but may instead be only semantically similar to motifs contained in the instances, a notion that subgraph-isomorphism–based membership does not capture. Secondly, we define the proxy membership in the classifier’s embedding space because we are concerned with how the classifier itself treats motifs and instances as similar or dissimilar, so that membership reflects the classifier’s embedding geometry rather than our own hand-crafted notion of the geometry of instances. Also, as you noted, computing subgraph isomorphism for $N$ instances and $E$ explanations would require $O(NE)$ calls to a subgraph matcher such as VF2, whose worst-case time complexity in the size of the graphs is exponential, whereas computing our proxy membership requires only $O(NE)$ distance computations on precomputed embeddings.

---

> > ### Author Response · Authors · 2025-11-23
> > **Response to Reviewer GqAd(Continued)**
> >
> > **W2.**  We agree that $M_r$ and $Y$ are strongly correlated. However, we do not claim otherwise. Quoting from the part in question, we only claim that "conditioned on the true membership $M^\star$, the residual noise in $M_r$ is independent of $Y$" which is what we need for Theorem 1 to be valid. Theorem 1 assumes that there exists a noise variable $\varepsilon$ such that $M_r = h(M^\star, \varepsilon)$ with $\varepsilon \perp Y \mid M^\star$. This means that, once the true membership $M^\star$ is known, the extra noise that makes $M_r$ differ from $M^\star$ does not contain additional information about $Y$. In other words, $M_r$ cannot have more information about $Y$ than $M^\star$ itself. In our setting, $M_r$ is a deterministic function of the embeddings and does not have access to any more information about labels than the true membership $M^*$ does satisfying the assumption in Theorem 1. We have revised the framing to make this more clear.
> >
> > **W3.**  We agree that this is a strong claim. It was primarily based on our own experience with discovery-based model-level explainers. However, we could not find explicit support for this statement in the existing literature, and since it is not in anyway related to our main contributions, we have decided to remove it from the paper.
> >
> > **W4.** In our setting, Greedy Gain Area (GGA) is computed on the coverage gain curve after the explainer has reached its maximal attainable coverage, and it measures how efficiently this coverage is accumulated as motifs are added in greedy order. A high GGA means that a small prefix of motifs already accounts for most of the coverage. In practice, this is desirable because it allows a user to understand the classifier’s behaviour by inspecting only a few motifs. Conversely, if the same coverage can only be achieved by adding many motifs, then understanding the model requires inspecting a larger set, which increases supervision cost.
> >
> > We agree that this does not by itself rule out redundancy among the remaining motifs. This is precisely why we introduce the Overlap metric, which separately quantifies how much the selected motifs cover the \emph{same} instances. In practice, we do not interpret GGA in isolation: high GGA is preferred only when it is accompanied by high Coverage and low Overlap; a set with high GGA but also high Overlap would be penalized by the latter as overly redundant. Thus, an explainer that optimizes all three metrics together under a fixed motif budget is naturally encouraged to find a small, diverse set of motifs that efficiently covers the classifier’s behaviour.
> >
> > **W5.**  We answer the question in two parts. Firstly, for more complex classification heads such as MLPs, the Lipschitz constant can be upper-bounded efficiently via spectral norms (e.g., by multiplying the spectral norms of the weight matrices). Our sufficiency bounds hold for any choice of radius at which Coverage is computed and any valid Lipschitz upper bound; only the tightness of the certificate depends on how tight this Lipschitz estimate is and how close the chosen radius is to the optimal one. Coverage is, by construction, a non-decreasing function of the radius. Overlap typically also grows as the radius increases, since each motif covers more graphs, whereas the behaviour of GGA as a function of the radius is less straightforward to characterize intuitively.
> >
> > Secondly, there is an alternative way to view more complex classifier heads such as MLPs that brings us back to the linear setting. Since our Lipschitz condition is required on the classifier head acting on the embedding space, we can absorb all but the last linear layer of the MLP into the embedding function and treat the final linear layer as the classification head. The embeddings are then taken to be the inputs to this last linear layer. Under this factorization, we are again in the regime where the Lipschitz constant of the head can be computed exactly as the spectral norm of a single linear layer, as done in the main paper, and our analysis with linear heads apply unchanged.
> >
> > **W6.**  We have revised the main text to make the problem setting as clear as possible and also added pointers to the appendix where necessary. We have also revised the Appendix to include all missing  implementational details and figures of how the motifs look like. Please see the corresponding revision in the main text and Appendix D for all the details.

---

> > > ### Author Response · Authors · 2025-11-23
> > > **Response to Reviewer GqAD(Continued)**
> > >
> > > **W7.** By evaluation, we meant calculation of the metric values for the explanation sets. For each class, the motifs have the same general shape but may have different number of constituent nodes. The number of nodes is sampled randomly within a predefined set range.  The graphs in a class and good explanation  set for that class contains instances of the motif generated from the same random motif generator.  In particular, the good set contains 5 instances of the motif generated from this random generator.  We have revised the main text of this part to clarify this.
> > >
> > > **W8.** You are indeed correct. This was a typo. We meant "near-zero" here. The whole description of this experiment has been updated accordingly.
> > >
> > > **W9.**  We thank the reviewer for pointing this out. We agree that, in the 4Shapes experiment, a plausible decision rule for Class 2 is exactly as you describe: the GNN can solve the task by recognizing three of the four motifs and treating Class 2 as a ``none of the above'' default when no other motif is present. Our additional experiments (now described in the text) indeed confirm that other  out-of-distribution graphs such as Erd\H{o}s--R\'enyi  graphs, which do not contain any of the other three motifs, are also predicted as Class~2 while the class specific motif receives lower class scores.
> > >
> > > However, we still believe this experiment highlights important complementary properties of Class score and Coverage. Inspecting Class score alone, would convince an user that the BA motif has been learnt by the classifier as evidence for Class 2 rather than absence. Only by inspecting Coverage, one sees that the BA motifs live far off the manifold of real instances labelled as Class 2 and inspecting class score of high coverage motifs of Class 2 shows that the classifier is not trigerred as well by the discriminative motif of Class 2. Hence, the absence of high scoring motifs with high coverage in this case together depicts that the classifier uses the decision rule based on absence of evidence on Class 2. This understanding also has important consequences in deciding whether the classifier should be used to identify instances of Class 2 in the wild. We have revised the experiment description to reflect this understanding.
> > >
> > > Finally, the fact that OOD pathological motifs often achieve high class score despite there being more in-distribution discriminative motifs that the classifier recognizes has been extensively discussed in prior-model level literature. Most prominently D4Explainer(NeurIPS 2023) discuss this phenomenon in their introduction itself and Graphon-Explainer(TMLR 2024) and GNNInterpreter(ICLR 2023) discuss that explanations such as isolated nodes on some target classes despite there being valid high scoring explanations.
> > >
> > > **W10.**  We have revised this part to make it clear.
> > >
> > > **W11.** We have updated the Github repo accordingly and include all the details in the ReadMe.

---

> ### Author Response · Authors · 2025-11-23
> **Response to Reviewer GqAD(Continued)**
>
> **Q1.**  Yes it would be zero but only when the explanation set $E_c$ is optimal in the sense that the classifier behaviour on the set of graphs labelled as class $c$ can be perfectly reconstructed from $E_c$ which is how sufficiency risk is characterized. In other words, an optimal $E_c$ must contain the motifs that the classifier consistently relies on for predicting the class identity on the graphs it has labelled as class $c$.
>
> **Q2.** The true membership function $M^\star$ used by the classifier remains unknown like we assume in the paper. Also  the proxy membership function $M_r$ is entirely defined on the embedding space. Therefore, it would depend on how the classifier embeds the instances in the class and the explanation motifs. So in the case you mention for GLGExplainer which builds explanations from concepts it has learned using the concept projection layer, $M_r$ would be defined on how the constructed explanations are embedded relative to embeddings of instances that the classifier has identified as belonging to the target class.
>
> **Q3.** We indeed thought about doing so. However, Coverage already captures a part of this information as can be seen as bad explanation sets get near zero coverage because they are farther away from embeddings of real instances in the target class while good explanation sets get much higher coverage owing to their proximity to class positive embeddings.
>
> **Q4.** Yes, we do believe this is a noteworthy issue. That is precisely why, our metrics are designed to reflect the embedding geometry. Infact, the farther away from the distribution the explanation motif is,  the less likely it is that the model behaves or decides on the motif the same way it decides on the distribution of real instances. That is why it is believe to find/generate explanation motifs that not only trigger the right class prediction but also lie close to the distribution of real instances. The latter part is exactly what metrics like Coverage measure.
>
> **Minors.** We have fixed the citation format, added the required reference and use "class score" uniformly. We will surely address **M1** and **M3** soon.
>
> **Thank You Note.**  We have found your review very constructive and it has really helped us to improve the quality of our manuscript. Regardless of the final decision, we highly appreciate your efforts, which we believe has made our manuscript better. We look forward to hearing from you on our responses and revisions.

---

> > ### Comment · Reviewer_GqAd · 2025-11-25
> > **Answer to Rebuttal**
> >
> > Thank you for your detailed clarifications. Here is a follow-up to some remaining concerns:
> >
> > **W0:**
> >
> > It would be nice to have this description in the paper, to clarify how you came up with this definition.
> >
> >
> > **W4:**
> >
> > > This is precisely why we introduce the Overlap metric, which separately quantifies how much the selected motifs cover the same instances.
> >
> > Thank you for the clarification. Nonetheless, I believe putting GGA in relation to Overlap may not rule out all possible failure cases of the GGA metric. Consider a toy example where the explanation consists of
> >
> > - A: a few high-quality motifs that appear across different samples, meaning it achieves high Coverage, high GGA, and low Overlap (to my understanding, therefore, the best scenario).
> >
> > - B: many other *noise* motifs that do not appear in any sample, and are spuriously generated by the explainer.
> >
> > Then, to my understanding, the final Coverage, GGA, and Overlap will have close to the best scores, meaning the set of explanatory motifs is considered to be of *good quality*. Can you please comment on this example? How does your metric penalize this scenario?
> >
> >
> > > In practice, we do not interpret GGA in isolation
> >
> > My feeling is that this could be a disadvantage of your suite of metrics, that is, it requires comparing metric values across different metrics with different semantics. In practice, it may not be easy to *calibrate* these metrics, and it may be hard to come to a unique conclusion or to a uniform ranking of explainers. Could you please comment on this?
> >
> >
> > **W9:**
> >
> > I personally found it surprising that, although the model does not recognize any motif for predicting class 2 (but rather relies on the absence), the BA motif live far off the manifold. Do you have some intuition for explaining this behaviour? I was expecting BA graphs to achieve high Coverage, as they should contain the minimal information the model needs to predict the correct label.
> >
> >
> > **Q3:**
> >
> > > We indeed thought about doing so. However, Coverage already captures a part of this information
> >
> > This sounds reasonable. Nonetheless, can you please clarify the pros of using Coverage over the raw distance?

---

> > > ### Author Response · Authors · 2025-11-26
> > > **Response to Remaining Concerns**
> > >
> > > Thank you for engaging with our work again. Please find our responses below:
> > > > **W0.** It would be nice to have this description in the paper, to clarify how you came up with this definition.
> > >
> > > **Response:** We have added it to the Appendix J in the revised paper. We also include the pointer to the appendix in the main text.
> > > >**W4(Part1)**         Thank you for the clarification. Nonetheless, I believe putting GGA in relation to Overlap may not rule out all possible failure cases of the GGA metric. Consider a toy example where the explanation consists of
> > > A: a few high-quality motifs that appear across different samples, meaning it achieves high Coverage, high GGA, and low Overlap (to my understanding, therefore, the best scenario).
> > > B: many other noise motifs that do not appear in any sample, and are spuriously generated by the explainer.
> > > Then, to my understanding, the final Coverage, GGA, and Overlap will have close to the best scores, meaning the set of explanatory motifs is considered to be of good quality. Can you please comment on this example? How does your metric penalize this scenario?
> > >
> > > We thank the reviewer for the thoughtful toy example. Under our current definitions, we do not explicitly penalize the case where an explainer outputs a high-quality subset $\mathbf{A}$ that already explains the class well, along with additional ``dead'' motifs $\mathbf{B}$ that never match any instance. This is because our metrics are designed to assess faithfulness/representativeness of the explanation whereas emitting extra motifs with no covered instances is primarily a wasted output budget issue rather than a failure of faithfulness.
> > >
> > > Importantly, **this corner case only arises when the noisy motifs have exactly zero support under the proxy membership**, i.e., $|S_k(r^\star)|=0$ (equivalently, they contribute zero marginal gain at the evaluation radius). If the additional motifs have nonzero support, then they necessarily affect at least one of the metrics: they either increase Coverage (if they cover new instances) or increase Overlap / alter the gain curve (if they mostly re-cover already covered instances), so the metric values will not remain identical.
> > >
> > > Moreover, the paper already contains a mechanism to address the reviewer’s scenario: the diagnostic stopping criterion based on gain-curve stagnation. In the proposed toy example, once the informative subset $\mathbf{A}$ has been selected, the greedy marginal gains for motifs in $\mathbf{B}$ are (near) zero, so the gain curve saturates and the stagnation diagnostic would indicate to an user that additional motifs cannot give more information about the classifier. So observing where the GGA curve stagnates not only identifies the motifs that give good coverage (case A) but  also gives the user the choice to ignore the noisy motifs (case B) that do not give more information.
> > >
> > > If desired, we can also add a lightweight auxiliary statistic to make this explicit, e.g., the fraction of motifs with zero support
> > > $\frac{1}{K}\sum_{k=1}^K \mathbb{1}\{|S_k(r^\star)|=0\}$
> > >  This directly quantifies ``wasted motifs'' without altering the core faithfulness metrics.
> > >
> > > Finally, we expect the reviewer’s corner case to be very rare in practice. An explainer that achieves high Coverage must already generate motifs that lie near the embedding region occupied by real instances. It is therefore unlikely that it simultaneously emits many additional motifs that are so off-manifold and OOD that they obtain **exactly zero support**.
> > >
> > > > **W4(Part2)**  My feeling is that this could be a disadvantage of your suite of metrics, that is, it requires comparing metric values across different metrics with different semantics. In practice, it may not be easy to calibrate these metrics, and it may be hard to come to a unique conclusion or to a uniform ranking of explainers. Could you please comment on this?
> > >
> > >  Actually this is not the case. We only look at GGA and Overlap when two explainers get equal Coverage as tiebreakers. If one explainer, gets greater Coverage, then it gives more information about the classifier and is superior. **Only in the case where, two explainers attain equal Coverage do we care about the other metrics**. GGA shows how efficiently the Coverage was attained. Overlap is primarily a diagnostic tool which has to be interpreted based on the Coverage attained by the explainer. High overlap with low coverage indicates mode collapse while high overlap with high coverage indicates that the embedding cloud itself maybe small therefore all motifs cover a large area of the embedding cloud. Thus, Coverage provides a clear primary ranking and the remaining metrics provide interpretable tie-breakers and diagnostics for different failure modes.

---

> > > > ### Author Response · Authors · 2025-11-26
> > > > **Responses to Additional Concerns(Continued)**
> > > >
> > > > >**W9**  I personally found it surprising that, although the model does not recognize any motif for predicting class 2 (but rather relies on the absence), the BA motif live far off the manifold. Do you have some intuition for explaining this behaviour? I was expecting BA graphs to achieve high Coverage, as they should contain the minimal information the model needs to predict the correct label.
> > > >
> > > > This is our intuition for why this happens. Since, the classifier decides based on "absence of evidence", the decision region for Class 2 is a large complement set in the embedding space. Everything that is not close to the other three motif regions can be mapped to Class 2 with high confidence. This region can extend far into low-density areas especially in the high dimensional embedding space where the model has seen little or no training data. So a BA graph can strongly trigger Class 2 prediction while being far from the embedding distribution of real Class-2 graphs (low coverage), because “not being like other classes” does not imply “being close to typical Class-2 samples". Also note that, a BA-only graph differs in global statistics  (motif missing; different node counts/degree profile; pooling magnitude effects) from a BA + Class 2 motif graph , so its embedding can land in a low-density region, even far away from where training samples of Class2 graphs are embedded though the classifier assigns it to Class 2.
> > > >
> > > > > **Q3** This sounds reasonable. Nonetheless, can you please clarify the pros of using Coverage over the raw distance?
> > > >
> > > >  Aside from the theoretical guarantees on Coverage and the metrics like GGA and Overlap that can be cleanly derived to produce additional diagnostics, there is a scenario which demonstrates the advantage clearly. The clear advantage is observed, not when a set of explanations is clearly OOD, but when it only covers part of the distribution. For instance in the MixedShapes dataset, where one explanation set contains motifs of single shapes and another contains motifs of both the shapes, the clear separation seen in the case of 4Shapes dataset is missing(see Figure 8b). Although the distributions of graph embedding to nearest motif embedding distances are clearly different for both the sets, they have a high degree of overlap unlike what is observed in 4Shapes. Hence, it is difficult to easily set a threshold distance.
> > > >     It might look from the figure(8b) that a threshold might be set based on the spread of the distributions. However, this is a fairly simple scenario. Imagine a scenario, where there are $n$ entities in a class and among them $k$ is covered by the explainer, then the distribution would look fairly complex making it a much more difficult problem.
> > > >
> > > > **We look forward to hearing your comments again after you have considered our response to the remaining concerns. Please let us know your thoughts on the content and quality of our manuscript and whether we could satisfactorily address all your concerns. Thank you.**

---

> > > > > ### Comment · Reviewer_GqAd · 2025-11-27
> > > > > **Answer to Rebuttal**
> > > > >
> > > > > Thank you once again for your detailed response.
> > > > >
> > > > > **W4 Part 1**
> > > > >
> > > > > I think there is a slight contradiction in the author's comment. First, they say:
> > > > >
> > > > > > Under our current definitions, we do not explicitly penalize the case where an explainer outputs a high-quality subset that already explains the class well, along with additional ``dead'' motifs that never match any instance
> > > > >
> > > > > But then
> > > > >
> > > > > > The paper already contains a mechanism to address the reviewer’s scenario: the diagnostic stopping criterion based on gain-curve stagnation
> > > > >
> > > > > Therefore, it is not clear to me whether the proposed metric can eventually penalize this case or not. Could you please clarify?
> > > > >
> > > > >
> > > > > **W4 Part 2**
> > > > >
> > > > > > Coverage provides a clear primary ranking and the remaining metrics provide interpretable tie-breakers and diagnostics for different failure modes
> > > > >
> > > > > Ok, this was not clear to me before. Thanks for the clarification.
> > > > >
> > > > > At this point, however, I believe that the example I outlined above can bring more severe consequences than the one outlined by the authors. In fact, authors have acknowledged that generating many many motifs can increase the Coverage, but it may decrease the other metrics. However, if the main ranking is done only based on Coverage, it means your ranking will favor explainers that generate many many motifs, and that may eventually contain ``dead'' motifs.
> > > > >
> > > > > Even if I agree that this case may not be very likely, I believe evaluation criteria should be robust even to corner cases. Also, generation-based explainers like XGNN are actually likely to generate unseen motifs, as they generate novel instances that are not guaranteed to be present in the dataset.
> > > > >
> > > > >
> > > > > **W9**
> > > > >
> > > > > Thank you for the clarification.
> > > > >
> > > > >
> > > > > **Q3**
> > > > >
> > > > > Thank you for the clarification.

---

> ### Author Response · Authors · 2025-11-27
> **Answers to Remaining Concerns**
>
> Thank you for your response to our rebuttal. Please find our responses to your questions below:
>
> > About the contradiction
>
> We apologize for the confusion. The clarification is:
>
> (i) The metrics do not penalize ``dead'' motifs.
> Under the definitions of Coverage/GGA/Overlap, motifs that never match any instance at the evaluation radius (i.e., $|S_k(r^\*)|=0$) contribute neither to the union nor to multiplicity, and they yield zero marginal gain in the greedy procedure. Therefore, the metric values themselves remain unchanged if such motifs are appended to an otherwise high-quality explanation set. In this sense, the metrics do not penalize this failure.
>
> (ii) The diagnostic stopping criterion addresses it operationally by allowing the user to identify which motifs add to the coverage and in what order.
> The gain-curve stagnation criterion detects when additional motifs provide negligible marginal coverage gain and thus cannot meaningfully improve the explanation. In the reviewer’s toy example, once the informative subset has been selected, the curve saturates and the diagnostic indicates to the user that the remaining motifs are non-contributory.
>
> If one wishes to penalize this explicitly, it is possible to incorporate a lightweight auxiliary statistic such as the fraction of motifs with zero support,
> $\frac{1}{K}\sum_{k=1}^K \mathbb{1}\{|S_k(r^\*)|=0\}$ to penalize this.
>
> > W4 Part 2
>
> First, ``dead'' motifs (zero support under the proxy membership) do not affect any of our metrics. If a motif covers no instance embeddings at the evaluation radius, i.e., $|S_k(r^\*)|=0$, then it contributes neither to Coverage (union), nor to Overlap (multiplicity), and yields zero marginal gain in the greedy gain curve (hence no effect on GGA). Therefore, appending such motifs cannot artificially improve Coverage or any other metric.
>
> Second, we agree that one can generate spurious motifs with small but nonzero support, could in principle inflate Coverage if we allowed arbitrarily large explanation sets. This is precisely why our comparison is budgeted. Explainers are evaluated on the same number of motifs $K$, consistent with the model-level goal of summarizing behaviour with a small set. $K$ can be chosen arbitrarily as any small integer. In our experiments, $K$ is chosen to be near the smallest value at which the gain curves of all the compared explainers have stagnated. Under this protocol, an explainer cannot ``game'' Coverage by emitting many extra motifs beyond saturation; those motifs are simply not part of the evaluated set. We will make this budgeted protocol more explicit in the revised manuscript.
>
> Finally, we emphasize that "unseen'' does not necessarily mean OOD. Generative explainers can produce novel motifs that are still in-distribution with respect to the classifier's embedding geometry. For example, on MUTAG, both XGNN and GNNInterpreter are generative; however, GNNInterpreter often yields disconnected graphs that are less representative of molecular structure, whereas XGNN tends to generate molecule-like motifs. Our metrics capture this distinction faithfully.
>
> **Please let us know again if we could address your concerns and your evaluation of our manuscript. We are eagerly waiting to hear back from you.**

---

### Official Review · Reviewer_JtxQ · 2025-10-28

**Soundness:** 3
**Presentation:** 3
**Contribution:** 3
**Rating:** 6
**Confidence:** 3

**Summary:**

This paper presents a principled evaluation framework for model-level explanations on graph neural networks (GNNs), introducing
theoretical guarantees via sufficiency risk and three novel metrics: Coverage, Greedy Gain Area (GGA), and Overlap. The work
addresses a critical gap in evaluating model-level explanations beyond classification scores, with rigorous theoretical foundations
and extensive experiments.

**Strengths:**

S1. The formalization of sufficiency risk provides a rigorous foundation for evaluating explanation quality. Theorems 1-3 and Proposition 1-2 establish certified bounds linking the proposed metrics to sufficiency risk, ensuring reliablity of metric even in a finite sample scenario.

S2. The three metrics offer complementary views of explanation quality (sufficiency, efficiency, non-redundancy), which reveals unfaithfulness, redundancy and mode collapse in the methods relying only on class score through extensive experiments.

S3. Experiments on 2 synthetic and 4 real-world datasets convincingly demonstrate the metrics' ability to expose limitations of class-score-based evaluation on model-level GNN explanations, such as pathological explanations and pattern collapse.

**Weaknesses:**

W1. The proxy function $M_r$ relies on nearest-neighbor distance in the embedding space. While simple and computationally efficient,
the paper doesn’t give the reason of choice and whether the measure is optimal for all GNN architectures or datasets or not.

W2. The description of “For each target class, an explanation set of ten motifs is generated by each explainer.” on page 8 indicates that
the current approach considers model-level explanations typically using small M(~10 motifs) in practice, as observed in the experiments. However, the computational complexity of GGA is related to O(NM^2) in Appendix F. A large explanation sets may lead to low scalability. Why does the paper define a ten motifs scenario in practice?

W3. Experiments only use GIN architectures (In Appendix D.2). The theoretical bounds depend on the Lipschitz constant of the
classifier head (In Theorem 2), which may vary with architecture (e.g., GAT or GCN), affecting metric stability.

**Questions:**

See the questions associated with Weaknesses

---

> ### Author Response · Authors · 2025-11-23
> **Response to Reviewer JtxQ**
>
> We are glad that you found our contribution to be addressing a **critical gap** in evaluation of model-level explanations. We are also happy that you found the **theoretical formalization rigorous and the experiments extensive**. Please find our responses to your questions pointwise below:
>
> **W1.** The primary reason we design the membership function on the embedding space like this is to respect the embedding geometry in a model-agnostic manner. Therefore, the membership naturally adapts to  variation in the embedding geometry with variance in GNN architectures is inherently taken care of by this design. Since we do not know the true membership function, the only measure of optimality of the proxy is being adaptable and computationally efficient which $M_r$ is by design.
>
> **W2.**     The number of motifs we used is dependent on when the Coverage of the explainer saturated . Empirically we found 10 to be a good number as generating more than 10 motifs did not result in any significant gain in Coverage. We would also like to point out that one of the central aims of model-level explanations is to gain an understanding of classifier behaviour in a manner that would require less human supervision than instance-level explanations. To achieve this model-level methods have always tried to summarize classifier behaviour on a target class using as less explanations are possible. This has been encouraged by designing means to generate more generalizable explanations which can provide insight into classifier behaviour on a large set of instances unlike instance-level explanations whose insights are instance specific and may not be valid across a broad range. Thus, in practice $M$ always remains very small for model-level explainers and scalability is not an issue in practice while evaluating GGA.
>
> **W3.** We have also performed experiments using GCN architectures on synthetic dataset whose description we include in the revised manuscript Appendix D.2. We would also like to point out that the metrics and the ordering of explainers should vary in certain scenarios. For example, changing the underlying classifier architecture may change which explainer is best suited for explaining the classifier model. Since our metrics are defined in the embedding geometry of the classifier they are bound to vary according to the explainer which produces the most sufficient explanations in the changed scenario. Therefore, rather than being an issue with stability we believe it is a strength of our metric framework. Further note that all the theoretical certificates continue to hold as it only depends on the Lipschitz property itself and the bounds remain tight as long as the Lipschitz constant can be calculated regardless of how the constant varies with architecture.
>
> **We thank you for your time you have invested in evaluating our contribution. Please let us know if you have any further questions/concerns regarding our manuscript. Alternatively, if you feel all your concerns are addressed, kindly consider strongly supporting our paper by a re-evaluation.**

---

> > ### Comment · Reviewer_JtxQ · 2025-11-28
> >
> > Thank you for the feedback. I will retain my previous score.

---

### Official Review · Reviewer_jLqD · 2025-10-29

**Soundness:** 2
**Presentation:** 2
**Contribution:** 2
**Rating:** 2
**Confidence:** 4

**Summary:**

This paper tackles the long-standing problem of evaluating model-level explanations for Graph Neural Networks (GNNs). Unlike instance-level explainers, which clarify decisions for individual samples, they focus on the explainers that aim to uncover class-discriminative motifs or graph patterns that the classifier consistently relies on. Current evaluation practices hinge almost entirely on target class scores, assuming high-scoring motifs faithfully represent the model’s reasoning. The authors argue this is flawed: high scores can result from pathological motifs detached from the true data distribution. To address this, they propose a theoretically grounded evaluation framework supported by distribution-free certificates that upper-bound an explanation’s insufficiency, or sufficiency risk.

**Strengths:**

1. This paper tackles a very important and long standing problem, namely, evaluating model-level GNN explanations.
2. The paper proposed three reasonable metrics.
3. The paper presented time analysis of the metrics.

**Weaknesses:**

1. The usage of the term "model-level" in this paper is not accurate. They develop metrics (Coverage, Greedy Gain Area, Overlap) that assume the explanation set is a collection of motifs (subgraphs) that represent class-discriminative patterns. However, many so-called “model-level explainer” methods do not output exactly “motifs” in this sense (i.e., compact sub-graph patterns extracted from or representative of the dataset). For example: XGNN generates class-specific instances (i.e. synthetic graphs) rather than mining motifs present in the training set. GCNeuron may produce Boolean rules or logic combinations rather than explicit graph motifs. The claim that their evaluation metrics are “for model-level explanations” is true only if the explainer produces motif-style explanations. If an explainer produces rules, logic formulas, or generated graphs that don’t align with “motifs from data”, then the metrics may not apply cleanly. The story told by the paper implies the metrics apply to all model‐level explainers, but they in fact only apply to motif‐extraction style ones. This needs to be clarified in the title, abstract, introduction, etc., instead of keeping it vague or overclaim.
2. Presentation needs improvement. Citation style needs to be corrected.
3. The three metrics, Coverage, GGA, and Overlap, are intuitive and could be justified directly from geometric reasoning. The introduction of sufficiency risk and distriThere is a fundamental mismatch between the proposed metrics and the explainers evaluated.
4. The metrics are designed for motif-extraction-based model-level explanations, where each explanation represents a class-discriminative subgraph sampled from or near the data distribution. However, the tested methods (XGNN, GNNInterpreter, PAGE) are generation-based and produce synthetic or prototype instances rather than extracted motifs. As a result, the evaluation does not truly measure what the proposed metrics are meant to assess. The reported results cannot substantiate the claimed effectiveness of the framework, since the metrics and the explainers operate on fundamentally different types of outputs.bution-free certificates adds complexity without offering real insight or practical necessity.

**Questions:**

N/A

---

> ### Author Response · Authors · 2025-11-18
> **Rebuttal to Reviewer jLqD**
>
> We appreciate that the reviewer thinks the paper tackles an important and longstanding problem in the field. We address each of the concerns that the reviewer raised topicwise  below:
> > **Scope of Model-Level Methods on which the Metrics Apply**
>
> >  The reviewer’s concern appears to stem from a misunderstanding of what we refer to as a *motif*. In our framework, a motif is defined as *any graph-structured explanation object that can be fed through the classifier to obtain its embedding*. Note that the term **motif** is not restricted to subgraphs in the model-level literature. Prior works such as Graphon-Explainer (TMLR 2024) and D4-Explainer (NeurIPS 2023), both generative model-level explainers, also refer to their generated explanations as motifs.
> Our proposed metrics operate entirely in the classifier’s embedding space and therefore do not require the explanation to be a mined subgraph from the dataset. The only requirement is that the explainer’s output is a graph that the GNN classifier can process.
>  We make this explicit in the **Preliminaries** section of the revised paper to clarify the scope of our metrics. Consequently, all baselines on which we conduct experiments whether generative, discovery-based, or prototype-based fall within the scope of our method. We hope this convinces the reviewer that we are not overclaiming our contribution and that the work treats the problem in full generality. We would be curious to hear the reviewer’s updated thoughts following this clarification.
>
> > **The three metrics, Coverage, GGA, and Overlap, are intuitive and could be justified directly from geometric reasoning. The introduction of sufficiency risk and distribution-free certificates adds complexity without offering real insight or practical necessity.**
>
> > We thank the reviewer for noting that Coverage, GGA and Overlap are geometrically intuitive. The purpose of introducing sufficiency risk and the distribution free certificates is precisely to show that these intuitive metrics have deeper theoretical consequences. While the geometric metrics quantify how an explanation set populates the classifier’s embedding space, the certificates establish what those geometric values mean for the behaviour of the classifier itself. In particular, the certificates translate metric values into statements of the form: “this explanation set accounts for at least a $(1-\epsilon)$ fraction of class $c$ with confidence $(1-\delta)$,” thereby providing a principled notion of explanation sufficiency. Thus, rather than adding unnecessary complexity, the certificates elevate the intuitive metrics into reliable, comparable and sample--aware quantities. They show that the geometric intuition underlying the metrics is not merely descriptive but connects directly to formal guarantees about classifier behaviour.
>
> > **Improvement in Presentation and Citation Formatting**
>
> >We will surely address the citation style in the next revision that we are preparing. We would also appreciate any particular comments regarding our presentation that the reviewer feels should be addressed, so that we can also take that into account in the upcoming revision.
>
> We believe we have addressed the major concerns raised by the reviewer regarding our manuscript. Please let us know if you have any further concerns in the meantime. Since you believe, we tackle an important and longstanding problem, kindly consider updating your score to support our paper so that our voice is heard at ICLR.
>
> **Please note we have also addressed the citation format in the latest revision as pointed out by the reviewer.**

---

> ### Author Response · Authors · 2025-11-25
> **Any Further Concerns/Comments from Reviewer jLqD**
>
> Dear Reviewer,
>
> As the rebuttal period is in its last week, we would like to kindly ask if you have any additional feedback for us to address.
>
> In our rebuttal, we clarified the specific point you raised regarding the meaning and role of **motifs** in our framework, ensuring that the terminology aligns with established usage in prior model-level explainability literature. Since this clarification directly resolves the concern upon which the evaluation was based, we hope the explanation has helped clear up the misunderstanding.
>
> If the clarification satisfactorily addresses your concern, we would be grateful if you could update your score accordingly. Otherwise, we would sincerely appreciate any remaining questions so that we can provide further explanation within the rebuttal timeline.
>
> Thank you again for your time and careful consideration.

---

### Official Review · Reviewer_qamU · 2025-11-03

**Soundness:** 2
**Presentation:** 3
**Contribution:** 2
**Rating:** 4
**Confidence:** 3

**Summary:**

This paper introduces three metrics, coverage, greedy gain area (GCA), and Overlap, to characterize the quality of the model-level explanations for GNNs. Coverage quantifies the likelihood a class is covered by a motif, determined by a threshold r and a distance measure (nearest motif). A tightness analysis verifies a suggested universal radius r for the sufficiency risk estimation.  GCA, based on coverage analysis, gives an area under the coverage curve. Overlap measures the possible redundancy of the motifs.

**Strengths:**

S1. Principled metric system and efficient assessment are important topics for graph learning.
S2. The paper provides an optimality analysis of parameter choices, such as r, and provides proper upper and lower bounds.
S3. The metrics are simple and generally applicable.

**Weaknesses:**

W1. There is a lack of time-cost analysis for the assessment process.
W2. Only the proposed metrics are measured; the methods lack more insightful justification from third-party measurements regarding consistency or any contradictory observations.
W3. It seems all or some of these measures are highly correlated. A necessary correlation or orthogonality analysis is needed.

**Questions:**

D1. It is not yet clear how the proposed metrics may encourage existing methods to seek computable optimal explanations under such measures, or whether they are conflicting or highly correlated, thereby indicating suboptimal yet Pareto-optimal solutions. An in-depth meta-measure analysis to verify potential trade-offs is needed. Other properties of metric systems, such as their susceptibility to scaling factors and user-defined preferences, and their invariance properties (will the value change significantly, suggesting different quality simply because a different distance function is used?) under the replacement of task-driven distance measures, deserve in-depth discussion.

D2. How the bound analysis helps in practice should be elaborated. It may suggest proper normalized measures—and indicate a fair assessment? The advantages of the proposed metrics, compared with other commonly adopted ones, should be discussed.

D3. How easy is it to compute such measures? What's the time cost? This needs further analysis.

D4. More insightful results could be summarized from Table 1. As is, only the results are shown, with little further analysis of how these measures help recommend or suggest promising solutions.

D5. More state-of-the-art GNN explainers should be compared and experimentally tested to show the benefit of the proposed measures.

---

> ### Author Response · Authors · 2025-11-18
> **Rebuttal to Reviewer qamU**
>
> We appreciate that the reviewer finds our contribution principled and our metrics simple and generally applicable. All the revisions are marked in **blue** in the revised version for ease of reference. Below we include a point by point response to each of the questions that the reviewer has raised:
>
> **D1**.     We have revised the paper to include a correlation analysis on the metrics( see Section 5.3 and Fig. 4). In summary, it reveals that there is no inherent tradeoff between these metrics and an explainer can be optimized to achieve optimal explanations by incentivizing it to: a) produce in-distribution explanations b) recognize distinct rationales that the classifier relies on for a target class. This is also supported by Figure 5 in the paper which explicitly shows that more indistribution explanation sets attain higher coverage.
>
> We have also included an in-depth discussion(in section 5.3) on the invariance properties of the metrics and robustness to geometric transformation including scaling. In short, the metrics remain stable under any transformation of the distance function that preserves relative distances between neighbors in the embedding space. Also, as Theorem 5 in the original paper showed we explicitly worked on a scale invariant notion of angular distance which is invariant to transformations such as normalization of embeddings.
>
> **D2.** Our paper introduces two complementary forms of guarantees: (i) asymptotic distribution free certificates, and (ii) finite sample statistical confidence intervals. The former provides a deeper consequence of what a given metric value means, specifically how sufficient an explanation set is and how extensively it covers the classifier reasoning modes. The latter quantifies statistical uncertainty and allows us to determine when observed differences in metric values are significant enough to conclude that one explanation method is strictly better than another for a given classifier.
>
> Regarding the advantages of our metrics over commonly used ones, the standard metric in model level explanation evaluation is the class score, which rewards explanations that yield a particular target prediction. Although class score is useful, it does not measure how many instances in a class are explained by a generated rationale and it does not penalize redundancy. An explainer may repeatedly produce nearly identical explanations and still obtain high class scores. As stated in the abstract, our metrics are designed to complement class score by rewarding explanations that are diverse, non redundant and aligned with in distribution reasoning patterns. Other metrics such as sparsity, although helpful for assessing explanation compactness, ignore the relationship between the classifier and the explanations, and therefore cannot assess model level sufficiency or representativeness. This distinction is already discussed in the Introduction and in the Related Work section on model level explanations.
>
> **D3.** We have already included a computational cost analysis in Appendix F. The measures are straightforward to compute. For any target class, the only model dependent step is to obtain the embeddings of the class positive instances and the embeddings of the explanation motifs, which requires a single forward pass through the GNN. All subsequent computations take place entirely in the embedding space. Coverage requires forming an $N \times M$ distance matrix and a minimum operation, Overlap is computed directly from the coverage sets, and GGA adds a small greedy selection over the $M$ motifs. As shown in Appendix~F, the overall worst case time complexity is $O(NM^{2})$, where $M$ is the number of motifs in an explanation set. In practice $M$ is very small (typically between $5$ and $10$), so the quadratic factor is negligible. Consequently, evaluating Coverage, GGA and Overlap is easily feasible.
>
> **D4.**     We have included a discussion of how the measures help recommend promising solutions in Section 5.3.  In short, finding in-distribution and diverse reasoning modes of the classifier gives good metric values and it is indeed possible to design explainers with such objectives since they do not conflict with each other as shown in the scatter plot of Figure 4. Please also see Section 5.2 where we have already included a brief intuition on how each method performs on each dataset and why.

---

> ### Author Response · Authors · 2025-11-18
> **Rebuttal to Reviewer qamU (Continued)**
>
> **D5.** We appreciate the suggestion and are in the process of running additional experiments with more state of the art model level explainers. We would like to clarify that the primary aim of this paper is not to benchmark explainers, but to introduce a principled evaluation framework for model level explanations. The paper already demonstrates that the proposed metrics capture complementary properties that are not reflected by the class score, which remains the most commonly used metric in prior work. Furthermore, we have evaluated three model level baselines that rely on three distinct mechanisms for generating or discovering explanations, tested across four diverse datasets spanning different scales and domains. We believe this already provides strong evidence of the practical usefulness and generality of the proposed measures. Benchmarking while very important in itself is orthogonal to the scope of this paper.
>
> We evaluated two additional recent model-level explainers: Gen-GraphEx(TNNLS 2025) which is based on a parametric probabilistic generative model with the parameters being estimated using a MLE and Graphon-Explainer(TMLR 2024) which uses graphons estimated from graphs classified to a target class as generative models. Note that we excluded the OGB-Molhiv dataset as we were unable to tune Gen-GraphEx  to get a reasonable performance and did not have the computational resources to compute the graphons for this dataset in the case of Graphon-Explainer. Here are the results:
>
> | Method               | Dataset        | Class         | Coverage      | GGA          | Overlap | Class Score    |
> |----------------------|----------------|---------------|---------------|--------------|---------|----------------|
> | **Gen-GraphEx**      | **MUTAG**      | Mutagenic     | 0.781±0.117   | 0.716±0.150  | 8.331   | 0.991±0.004    |
> |                      |                | Non-mutagenic | 0.775±0.185   | 0.682±0.235  | 6.741   | 0.989±0.003    |
> |                      | **Reddit-B**   | Class 0       | 0.899±0.051   | 0.664±0.065  | 5.712   | 1.000±0.000    |
> |                      |                | Class 1       | 0.883±0.038   | 0.785±0.048  | 6.518   | 1.000±0.000    |
> |                      | **IMDB-Multi** | Class 0       | 0.963±0.018   | 0.913±0.061  | 8.488   | 0.988±0.001    |
> |                      |                | Class 1       | 0.902±0.077   | 0.852±0.092  | 8.821   | 0.977±0.000    |
> |                      |                | Class 2       | 0.931±0.062   | 0.871±0.074  | 8.801   | 0.975±0.001    |
> | **Graphon-Explainer**| **MUTAG**      | Mutagenic     | 0.821±0.117   | 0.801±0.150  | 8.111   | 0.984±0.001    |
> |                      |                | Non-mutagenic | 0.858±0.185   | 0.788±0.235  | 7.961   | 1.000±0.000    |
> |                      | **Reddit-B**   | Class 0       | 0.885±0.051   | 0.711±0.065  | 7.017   | 0.983±0.001    |
> |                      |                | Class 1       | 0.831±0.038   | 0.774±0.048  | 7.181   | 0.984±0.000    |
> |                      | **IMDB-Multi** | Class 0       | 0.903±0.018   | 0.903±0.061  | 9.000   | 1.000±0.000    |
> |                      |                | Class 1       | 0.949±0.077   | 0.906±0.061  | 8.966   | 0.995±0.000    |
> |                      |                | Class 2       | 0.988±0.062   | 0.988±0.074  | 9.000   | 1.000±0.000    |
>
> The suprising result is that Gen-GraphEx achieves the best coverage along with perfect class scores on the Reddit dataset showing that the distribution of training graphs was captured by the generative model very well and the classifier recognizes in-distribution discriminative motifs that were missed by other models. The other trends remain the same as before. Namely, on IMDB-Multi all the explainers achieve high coverage with high GGA showing that the embedding cloud is compact and lesser number of motifs are needed to cover it. On MUTAG, they achieve comparable performance to XGNN and PAGE showing that they are better at capturing in-distribution motifs than GNNIntepreter.
>
> **Kindly let us know if you have any additional concerns. We believe we could address the concerns you had raised satisfactorily and they have significantly improved the quality of our manuscript. Please consider re-evaluating and supporting our paper if we indeed managed to alleviate your concerns.**

---

> ### Author Response · Authors · 2025-11-27
> **Friendly Reminder**
>
> Dear Reviewer qamU,
>
> Please let us know if you found the time to consider our responses and revisions and if you have any additional concerns. We believe your suggestion especially on the **meta measure analysis** has significantly strenghthened our paper and we are grateful for your constructive remarks. We are eagerly waiting for your feedback on the revised manuscript.

---

### Author Response · Authors · 2025-12-04
**A Summary of the Reviews and our Responses to them**

We thank the reviewers for their time and effort. As today is the last day on which the authors can comment on their submission, we summarise the reviews and our responses (and corresponding revisions) for the Chairs and for future readers of this page. Note that the OpenReview security incident unfortunately terminated ongoing discussions mid-way and did not allow reviewers to update their scores.

## TL;DR of reviews + how we addressed them

### Positive Points
1) **Timely and important problem.** Reviewer **jLqD** highlighted that the paper tackles an “**important and longstanding problem**” in evaluating model-level GNN explanations. Reviewer **GqAd** similarly noted that the work fills a “**big gap**” in the literature and has the "**potential to be a cornerstone paper**".

2) **Principled foundation with formal guarantees.** Reviewer **JtxQ** emphasized the rigorous theoretical grounding via **sufficiency risk** and valued the certified links between the proposed metrics and sufficiency risk (including reliability under finite samples).

3) **Simple, general, and complementary metrics.** Reviewer **qamU** appreciated that the metric system is “**simple and generally applicable**,” while Reviewer **JtxQ** noted that the metrics provide complementary views beyond class score (e.g., redundancy and coverage effects).

4) **Strong empirical demonstration.** Reviewer **JtxQ** found that the synthetic + real-world experiments convincingly show why class-score-only evaluation is insufficient and how Coverage/GGA/Overlap reveal distinctions it misses.

---

### Main Criticisms and How We Addressed Them

1) **Reviewer jLqD (Scope of the Metrics).**
**Concern:** The reviewer’s main criticism rested on a restrictive interpretation that “motifs” must be mined subgraphs, and therefore our evaluation would not apply to other model-level explainer outputs.
**Response/Revisions:** We clarified early (and revised the manuscript accordingly) that in our framework a “motif” means **any graph-structured explanation object that can be processed by the classifier to obtain an embedding** (including generated/prototype explanations), since the evaluation is defined in the classifier’s embedding space.
**Note:** We addressed this point very early (18th Nov) in the discussion period, **Reviewer jLqD did not respond further despite our followups**. We consider this criticism **wrong and non-representative of the content of the paper**.

2) **Reviewer GqAd**
**Concern:** The reviewer raised questions about definitions/assumptions and some experimental clarity, and indicated they would consider updating their score if the concerns were addressed.
**Response/Revisions:** The discussion indicates that these concerns **were addressed through clarifications and revisions**, and the reviewer acknowledged several clarifications during the exchange. However, the OpenReview incident blocked the discussion in the middle, preventing a clean closure. **The reviewer's critique was very constructive and significantly improved our manuscript.**

3) **Reviewer JtxQ (proxy choice, scalability, architecture dependence).**
**Concern:** Questions about using embedding-distance membership vs. graph-isomorphism-style criteria, scalability of the metrics, and dependence on the underlying GNN architecture.
**Response/Revisions:** We clarified that embedding-based membership reflects the classifier’s notion of similarity rather than a human's notio; we also clarified the computational profile (after embeddings are computed, operations are efficient in embedding space) and discussed how architecture dependence is expected and handled within the certified framework.

4) **Reviewer qamU (time-cost + meta-measure/robustness concerns).**
**Concern:** Requested  time-cost analysis and asked about correlation/orthogonality and robustness of the metrics.
**Response/Revisions:** We pointed the reviewer to the computational-cost explanation which already existed in the Appendix and conducted a detailed meta measure analysis on how the metrics behave relative to each other. We also expanded the discussion on robustness and highlighted how our metrics complement class score.
---

### Bottom line
All reviewers agree the paper addresses an important gap and introduces a principled evaluation framework beyond class score.  We undertook significant revisions and addressed each of the reviewers' comments in detail. We hope the Chairs  will weigh this updated context, particularly given that the ongoing discussion was interrupted by the platform incident.

---

### Meta-Review · Area_Chair_Eq6V · 2025-12-10

**Summary:**

This paper addresses the evaluation of model-level explanations for GNNs, which aim to identify class-discriminative motifs representing how a classifier recognizes target classes. The authors argue that the standard evaluation metric (class score) is insufficient, as high-scoring explanations may be pathological or fail to capture the full range of classifier reasoning. They first formally define the sufficiency risk for explanation adequacy. Based on that, they propose three metrics: Coverage (sufficiency), GGA (efficiency), and Overlap (redundancy).

**Reviewer Concerns:**

Initially, the reviewers raised significant concerns including clarity of the theoretical framework's necessity, scope of applicability, experimental presentation, and metric interpretation. Many concerns are largely addressed by the authors but with the following significant ones.

1. Architecture dependence.
No comprehensive empirical validation on complex non-linear heads. They authors only provide theoretical analysis.

2. Potential gaming of metrics (dead motifs)
This limitation may affect the robustness of these metrics.

3. Reliability of embeddings φ(Mₖ) for OOD motifs
This is a deep theoretical concern about the method's foundational assumption. The paper assumes embedding geometry is more stable/reliable than final class scores for OOD inputs, (classification score is affected by OOD, but the authors use the embedding function, part of the classifier, without consider OOD). They didn't provides theoretical or empirical evidence for this specific undrlying claim.

4. No actionable guidance
Missing pseudocode, step-by-step protocol, interpretation thresholds, worked examples. For an evaluation tool paper, this significantly limits usability.

Despite these limitations, the paper's core contribution, introducing the first theoretically-grounded suite of certified metrics for model-level explanation evaluation, is substantial and timely.  I recommend acceptance, but I strongly suggest the authors address the above concerns in the camera ready version.

**Reviewer Scores:**

qamU (4->6): Concerns largely addressed, would likely raise score.
 JLqD (2->4): Core definition clarified, but lack of engagement suggests modest increase.
 JtxQ (6->6): Positive throughout, confirmed unchanged score.
 GqAd (2->4): Detailed concerns largely met, but some nuanced issues remain, leading to borderline score.

---

### Decision · Program_Chairs · 2026-01-26

Accept (Poster)